# Extreme Blind Image Restoration via Prompt-Conditioned Information Bottleneck

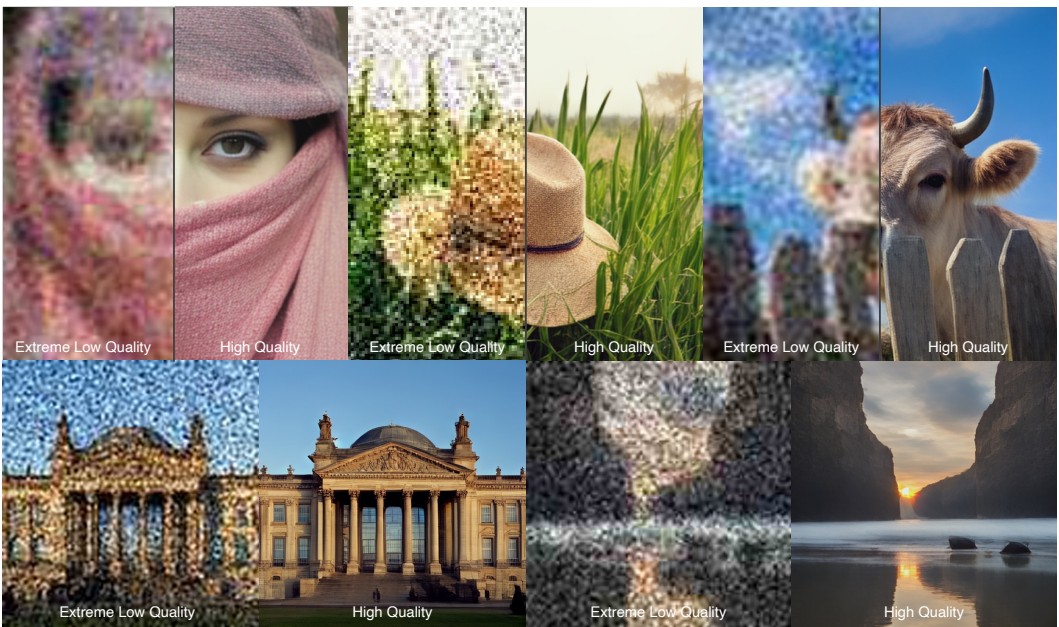

Figure 1: Extreme blind image restoration with our Prompt-Conditioned Information Bottleneck method for various fully blind images. Our method produces high-quality results with fine detail.

## ABSTRACT

Blind Image Restoration (BIR) methods have achieved remarkable success but falter when faced with Extreme Blind Image Restoration (EBIR), where inputs suffer from severe, compounded degradations beyond their training scope. Directly learning a mapping from extremely low-quality (ELQ) to high-quality (HQ) images is challenging due to the massive domain gap, often leading to unnatural artifacts and loss of detail. To address this, we propose a novel framework that decomposes the highly challenging ELQ-to-HQ restoration process. We first learn a projector that maps an ELQ image onto an intermediate, less-degraded LQ manifold. This intermediate image is then restored to HQ using a frozen, off-the-shelf BIR model. Our approach is grounded in information theory; we provide a novel perspective of image restoration as an *Information Bottleneck* problem and derive a theoretically-driven objective to train our projector. This loss function effectively stabilizes training by balancing a low-quality reconstruction term with a high-quality prior-matching term. Our framework enables *Look Forward Once* (LFO) for inference-time prompt refinement, and supports plug-and-play strengthening of existing image restoration models without need for finetuning. Extensive experiments under severe degradation regimes provide a thorough analysis of the effectiveness of our work.

# 1 INTRODUCTION

Image restoration is a long-standing problem in computer vision that focuses on recovering a clean high-quality (HQ) image from its degraded low-quality (LQ) observation. Denoting the space of realistic HQ images as $\mathcal{X}_{\text{HQ}}$ and the space of LQ images as $\mathcal{X}_{\text{LQ}}$, the objective is to learn the mapping,

$$\mathcal{R} : \mathcal{X}_{\text{LQ}} \rightarrow \mathcal{X}_{\text{HQ}} \tag{1}$$

In recent years, deep learning-based methods have achieved remarkable success in this domain, replacing traditional model-based approaches by learning powerful image priors directly from large-scale datasets (Zhang et al., 2017; Zamir et al., 2022; Wang et al., 2022; Liang et al., 2021; Dong et al., 2014; Chen et al., 2023). However, these methods often assume a simple, known degradation process, which is impractical for real-world scenarios where degradations are a complex combination of various factors (e.g., compression artifacts, sensor noise, motion blur, downsampling, etc). Such *blind* nature of real-world scenarios makes the already ill-posed problem of restoration significantly more difficult, as a single LQ image could theoretically result from numerous combinations of clean images and degradation functions (Wang et al., 2021; Zhang et al., 2021).

Blind Image Restoration (BIR) tackles this challenge directly by aiming to recover a high-quality, clean image from a low-quality counterpart without prior knowledge of the degradation. Recent methods in BIR (Chihaoui et al., 2024; Chihaoui & Favaro, 2025; Xiao et al., 2024; Lin et al., 2024) demonstrate impressive performance by modeling a rich and randomized space of synthetic degradations to train robust restoration networks. Formally, the mapping from $\mathcal{X}_{\text{HQ}}$ to $\mathcal{X}_{\text{LQ}}$ is set as,

$$\mathcal{D} : \mathcal{X}_{\text{HQ}} \rightarrow \mathcal{X}_{\text{LQ}} \tag{2}$$

where $\mathcal{D}$ is modeled as a compounded function of random degradations. However, existing methods still face significant challenges when confronted with Extreme Blind Image Restoration (EBIR), where the input image suffers from exceptionally severe and compounded degradations *beyond the scales of their original training settings*.

In EBIR, the combined effect of severe degradation and their blind, random composition causes the domain gap between $\mathcal{X}_{\text{HQ}}$ and the space of ELQ images $\mathcal{X}_{\text{ELQ}}$ to become *massive*. Denoting the mapping from $\mathcal{X}_{\text{ELQ}}$ to $\mathcal{X}_{\text{HQ}}$ as follows,

$$\mathcal{R}_E : \mathcal{X}_{\text{ELQ}} \rightarrow \mathcal{X}_{\text{HQ}} \tag{3}$$

A trivial approach such as learning $\mathcal{R}_E$ end-to-end with a single model would lead to suboptimal results due to the highly complex transformation and considerably large solution space. The network would struggle in learning a stable and effective mapping, producing outputs with unnatural textures or failing to recover essential structural details. To address this challenge, we factorize the mapping $\mathcal{R}_E$ into two simpler sub-problems. Specifically, instead of learning $\mathcal{R}_E$ directly, we first project an ELQ image onto an intermediate LQ manifold via a trainable projector $f_\theta : \mathcal{X}_{\text{ELQ}} \rightarrow \mathcal{X}_{\text{LQ}}$, and then apply a frozen pretrained restoration model $g : \mathcal{X}_{\text{LQ}} \rightarrow \mathcal{X}_{\text{HQ}}$. Such decomposition of $\mathcal{R}_E$ into a $\mathcal{X}_{\text{ELQ}} \rightarrow \mathcal{X}_{\text{LQ}} \rightarrow \mathcal{X}_{\text{HQ}}$ path effectively shrinks the solution space via an intermediate distribution and stabilizes learning in the severe, compound-degradation regime.

We ground the design of our method by a novel perspective of the image restoration task as an **Information Bottleneck (IB)** problem (Tishby et al., 2000; Tishby & Zaslavsky, 2015; Alemi et al., 2016), originally designed for the *compression* of one random variable while maintaining the *relevance* of another. We reframe the task of image restoration by introducing a *degrade-back* simulation for $\mathcal{X}_{\text{LQ}} \rightarrow \mathcal{X}_{\text{HQ}} \dashrightarrow \mathcal{X}_{\text{LQ}}$, and regarding a reconstructed HQ image as a *compressed* form of the given LQ image in terms of degradation and artifacts. Importantly, this perspective provides a theoretically-driven objective consisting of an LQ reconstruction term and an HQ prior-matching term. Formally, we derive the *Image Restoration Information Bottleneck (IRIB) loss* and its variant for training $f_\theta$ to restore ELQ images with a fixed pretrained $\mathcal{X}_{\text{LQ}} \rightarrow \mathcal{X}_{\text{HQ}}$ mapping network $g$.

The proposed decomposition of the IR task also enables application of auxiliary refinement methods during inference time. Our pipeline exposes an intermediate space $\mathcal{X}_{\text{LQ}}$ which serves as an appropriate location for refinement. Specifically, we introduce **L**ook **F**orward **O**nce (**LFO**)[1] to refine conditioning (*e.g.*, text prompts) using intermediate samples before finally mapping to an HQ

---

[1] Name inspired by the work (Zhang et al., 2022).

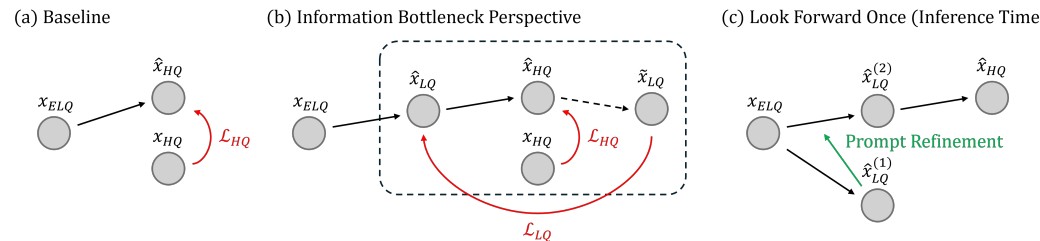

Figure 2: **(a) Baseline Methods:** Conventional restoration learns a single mapping $\mathcal{X}_{\text{ELQ}} \to \mathcal{X}_{\text{HQ}}$ from Extreme LQ images to HQ images. This is a highly ill-posed problem, thus training with only the HQ matching objective $\mathcal{L}_{\text{HQ}}$ becomes highly impractical. **(b)** Our **Information Bottleneck Perspective** introduces an intermediate distribution $\mathcal{X}_{\text{LQ}}$ and simulates a *degrade-back* channel, leading to an additional LQ reconstruction objective $\mathcal{L}_{\text{LQ}}$. **(c) Look Forward Once (LFO):** The decomposition of the $\mathcal{X}_{\text{ELQ}} \to \mathcal{X}_{\text{HQ}}$ problem to $\mathcal{X}_{\text{ELQ}} \to \mathcal{X}_{\text{LQ}} \to \mathcal{X}_{\text{HQ}}$ allows for auxiliary refinement in the intermediate LQ distribution during inference time. Prompt refinement via LFO improves performance of extreme image restoration by finding a finer image in the LQ domain.

image. This method can be used *iteratively* to enhance performance during inference time with minimal overhead. Furthermore, a well-trained projection $f_\theta : \mathcal{X}_{\text{ELQ}} \to \mathcal{X}_{\text{LQ}}$ would allow us to leverage $f_\theta$ to enhance the restoration properties of *any* pretrained image restoration model $g : \mathcal{X}_{\text{LQ}} \to \mathcal{X}_{\text{HQ}}$, provided that the subspace $\mathcal{X}_{\text{LQ}}$ is identical. In fact, many state-of-the-art IR methods are trained with the same degradation $\mathcal{D}$; our method proves effective for broadening the restoration capabilities of pretrained BIR models up to *extreme* BIR in a plug-and-play fashion without training.

In summary, our contributions are as follows:

- We recast extreme blind image restoration as an Information Bottleneck (IB) problem and derive the Image Restoration Information Bottleneck (IRIB) objective that couples an LQ reconstruction term with an HQ prior-matching term.

- We introduce a decomposition of the Extreme Blind Image Restoration problem, and an efficient method for solving it leveraging the IRIB objective. Such decomposition shrinks the solution space, enabling efficient training while producing realistic high-quality results.

- Our framework enables Look Forward Once, a strategy for inference-time prompt refinement that can be iteratively applied, and plug-and-play strengthening of existing image restoration models without need for finetuning.

## 2 RELATED WORK

**Blind Image Restoration.** Blind image restoration (BIR) aims to restore images degraded by $\mathcal{D}_\phi$ without knowledge of $\phi$. In real-world scenarios, we do not have knowledge of both $\mathcal{D}$ and $\phi$, causing the IR task to be *fully blind*. Although solving fully blind IR allows for more practical use cases, the intensified ill-posedness induces additional difficulties. Random-degradation training has become a practical recipe for solving fully blind IR: BSRGAN (Zhang et al., 2021) designs a shuffled, multi-stage degradation pipeline to synthesize diverse low-quality inputs for training, and Real-ESRGAN (Wang et al., 2021) incorporates higher-order degradations. By virtue of such powerful training pipelines, recent BIR models are now capable of producing remarkable results (Wu et al., 2024b; Wang et al., 2024a; Sun et al., 2025; Wu et al., 2024a; Lin et al., 2024). Among them, DiffBIR (Lin et al., 2024) casts blind restoration as a two-stage pipeline by conditioning a latent diffusion prior with region-adaptive guidance. SeeSR (Wu et al., 2024b) uses semantic prompts to restore images, but its diffusion-based nature demands high inference time. OSEDiff (Wu et al., 2024a) replaces multi-step diffusion sampling with a one-step diffusion-based generator stabilized by variational score distillation. A major challenge for all pretrained BIR models is that, once trained, *they cannot be leveraged to restore images of degradations beyond their training regime*. This is a critical drawback, as the additional training incurred is non-trivial due to the severe ill-posedness of the problem. Thus, we propose to *decompose* the image restoration process to mitigate the ill-posedness by introducing an intermediate distribution of less degradation. Leveraging a theoretically driven loss, our method enables successful restoration even for extremely degraded images.

**Decomposition of Image Restoration.** A long line of image restoration methods reduces ill-posedness by inserting intermediate states that progressively constrain the solution. Pyramidal designs reconstruct at multiple resolutions (*e.g.*, progressive Laplacian pyramid of LapSRN (Lai et al., 2017), multi-scale deblurring via coarse-to-fine updates (Nah et al., 2017)), and CinCGAN (Yuan et al., 2018) bridges to a cleaner LR domain before upscaling. GAN-prior methods such as GLEAN (Chan et al., 2021) and PULSE (Menon et al., 2020) search or bank latent codes as an intermediate manifold. Recent two-stage pipelines (DiffBIR (Lin et al., 2024), SeeSR (Wu et al., 2024b)) first move inputs to a cleaner intermediate (degradation-removed or semantics-aligned) state, then invoke a powerful diffusion prior for detail synthesis. Chain-of-Zoom (Kim et al., 2025) attempts a similar decomposition of the super-resolution task for extreme super-resolution during inference time. However, these methods do not solve the problem of restoring from extremely low-quality inputs under compounded, unknown degradations, where learning a direct mapping to high quality remains ill-posed. Instead of proxy scales/latents or denoise-then-regenerate stages, our method inserts a low-quality intermediate distribution aligned with real degradations and trains a projection module to populate it while the restoration backbone remains frozen.

## 3 PRELIMINARIES

**The Information Bottleneck.** Let $(X, Y) \sim p(x, y)$. The Information Bottleneck (IB) principle seeks a stochastic representation $Z$ of $X$ that keeps only task-relevant information about $Y$ while compressing $X$ (Tishby et al., 2000; Tishby & Zaslavsky, 2015). The IB seeks to minimize

$$\min_{p(z|x)} \mathcal{L}_{\text{IB}} := I(X; Z) - \beta I(Z; Y), \quad \beta > 0. \tag{4}$$

Minimizing the compression term $I(X; Z)$ enforces compression, forcing the representation $Z$ to discard information irrelevant to the task. Maximizing the relevance term $I(Z; Y)$ ensures that the representation $Z$ remains useful for predicting the target $Y$.

**The Variational Information Bottleneck.** A primary obstacle to the direct application of the IB principle in deep learning is the computational intractability of mutual information. The Deep Variational Information Bottleneck (VIB) (Alemi et al., 2016) provides a practical solution by deriving a tractable variational lower bound on the IB objective. This bound can be readily optimized using standard stochastic gradient-based methods. Formally, a parametric encoder $q_\phi(z \mid x)$, decoder $q_\psi(y \mid z)$, and a simple variational prior $r(z)$ (*e.g.*, $\mathcal{N}(0, I)$) is introduced to bound the two mutual information terms $I(X; Z)$ and $I(Z; Y)$.

An upper bound on $I(X; Z)$ is derived as follows:

$$\begin{aligned} I(X; Z) &= \mathbb{E}_{p(x,z)} \left[ \log q_\phi(z|x) - \log p(z) \right] \\ &\leq \mathbb{E}_{p(x,z)} \left[ \log q_\phi(z|x) - \log r(z) \right] = \mathbb{E}_{p(x)} \left[ \text{KL}(q_\phi(z|x) \parallel r(z)) \right] \end{aligned} \tag{5}$$

where the inequality comes from $\text{KL}(p(z) \parallel r(z)) \geq 0 \Rightarrow \mathbb{E} \log p(z) \geq \mathbb{E} \log r(z)$. Similarly, a lower bound on $I(Z; Y)$ is derived as follows:

$$I(Z; Y) = \mathbb{E}_{p(y,z)} \left[ \log p(y|z) \right] - \mathbb{E}_{p(y)} \left[ \log p(y) \right] \geq \mathbb{E}_{p(x,y)} \mathbb{E}_{q_\phi(z|x)} \left[ \log q_\psi(y|z) \right] + H(Y), \tag{6}$$

where the inequality comes from $\text{KL}(p(y|z) \parallel q_\psi(y|z)) \geq 0 \Rightarrow \mathbb{E} \log p(y|z) \geq \mathbb{E} \log q_\psi(y|z)$. The $H(Y)$ term is constant with respect to model parameters in supervised setups. Combining the bounds in the IB Lagrangian of Eq. 4, the following holds:

$$\mathcal{L}_{\text{IB}} \leq \underbrace{\mathbb{E}_{p(x)} \left[ \text{KL}(q_\phi(z|x) \parallel r(z)) \right]}_{\text{compression term}} - \beta \underbrace{\mathbb{E}_{p(x,y)} \mathbb{E}_{q_\phi(z|x)} \left[ \log q_\psi(y|z) \right]}_{\text{relevance term}} + \text{const.} \tag{7}$$

Thus, minimizing the upper bound of Eq. (7) is equivalent to minimizing the VIB loss

$$\mathcal{L}_{\text{VIB}}(\phi, \psi) = \mathbb{E}_{p(x,y)} \left[ \mathbb{E}_{q_\phi(z|x)} \left[ -\log q_\psi(y|z) \right] + \beta \, \text{KL}(q_\phi(z|x) \parallel r(z)) \right]. \tag{8}$$

**Connection to $\beta$-VAE.** Setting the IB relevant variable to the input itself ($Y = X$; self-prediction) and using the same variational family as in VIB, we obtain the $\beta$-VAE loss:

$$\mathcal{L}_{\beta\text{-VAE}}(\phi, \psi) = \mathbb{E}_{p(x)} \left[ \mathbb{E}_{q_\phi(z|x)} \left[ -\log q_\psi(x|z) \right] + \beta \, \text{KL}(q_\phi(z|x) \parallel r(z)) \right]. \tag{9}$$

In this work, we redefine the problem of image restoration in the lens of the Information Bottleneck principle, and propose a theoretically-driven method for effectively solving extreme image restoration with pretrained image restoration models.

## 4 EXTREME IMAGE RESTORATION AS AN INFORMATION BOTTLENECK

### 4.1 OVERVIEW

Let $X \sim p_{\text{LQ}}$ be low-quality (LQ) images and $Z \sim p_{\text{HQ}}$ be high-quality (HQ) images. Baseline methods for image restoration aim to find an effective mapping $g : \mathcal{X}_{\text{LQ}} \to \mathcal{X}_{\text{HQ}}$:

$$\underbrace{X}_{\text{LQ}} \xrightarrow{g} \underbrace{Z}_{\text{HQ proxy}} , \qquad \text{with } Z \in \mathcal{X}_{\text{HQ}}. \tag{10}$$

In this work, we present a novel perspective of the image restoration problem as an Information Bottleneck (IB) problem via simulating the *degrade-back* of an HQ proxy:

$$\underbrace{X}_{\text{LQ}} \xrightarrow{g} \underbrace{Z}_{\text{HQ proxy}} \xdashrightarrow{d_\psi} \tilde{X}, \qquad \text{with } Z \in \mathcal{X}_{\text{HQ}}, \tilde{X} \in \mathcal{X}_{\text{LQ}}. \tag{11}$$

where $d_\psi : \mathcal{X}_{\text{HQ}} \to \mathcal{X}_{\text{LQ}}$ is a possible degradation channel. This shows strong correspondence to the self-prediction case (*i.e.*, $\beta$-VAE) of the Variational Information Bottleneck, allowing us to directly leverage the $\beta$-VAE loss of Eq. (9) to formulate the **Image Restoration Information Bottleneck (IRIB)** loss:

$$\mathcal{L}_{\text{IRIB}}(\phi, \psi) = \mathbb{E}_{p(x)} \Big[ \underbrace{\mathbb{E}_{q_\phi(z|x)} \left[ -\log q_\psi(x|z) \right]}_{\mathcal{L}_{\text{LQ-recon}}} + \beta \underbrace{\text{KL}(q_\phi(z|x) \parallel r(z))}_{\mathcal{L}_{\text{HQ-prior}}} \Big]. \tag{12}$$

where $q_\psi(x|z)$ parameterizes the LQ-likelihood induced by $d_\psi$. Note that this loss mirrors that of $\beta$-VAE exactly: a reconstruction term and a prior matching term.

**Information Bottleneck for Extreme Image Restoration.** Given the loss $\mathcal{L}_{\text{IRIB}}$, we now propose a novel method for using it to solve the problem of extreme image restoration. Specifically, we aim to leverage a *pretrained* mapping $g$ and leverage it to solve restoration tasks for extremely low-quality images that are degraded beyond the training configuration of $g$. Let $X^{(E)} \sim p_{\text{ELQ}}$ be extremely low-quality (ELQ) images. We aim to find an effective mapping $f_\theta : \mathcal{X}_{\text{ELQ}} \to \mathcal{X}_{\text{LQ}}$:

$$\underbrace{X^{(E)}}_{\text{ELQ}} \xrightarrow{f_\theta} \underbrace{\hat{X}}_{\text{LQ proxy}} \xrightarrow{g} \underbrace{\hat{Z}}_{\text{HQ proxy}} \xdashrightarrow{d_\psi} \tilde{X}, \qquad \text{with } \hat{Z} \in \mathcal{X}_{\text{HQ}}, \hat{X} \in \mathcal{X}_{\text{LQ}}, \tilde{X} \in \mathcal{X}_{\text{LQ}}. \tag{13}$$

Since a naive mapping to directly learn $\mathcal{X}_{\text{ELQ}} \to \mathcal{X}_{\text{HQ}}$ would impose impractical training costs and suboptimal results, we leverage the proposed IRIB loss *to introduce an LQ proxy* for restricting the solution space in an intermediate distribution. The IRIB loss of Eq. (12) thus becomes:

$$\mathcal{L}_{\text{IRIB-ELQ}}(\theta; \phi, \psi)$$
$$= \mathbb{E}_{(x^{(E)}, x) \sim p_{\text{ELQ,LQ}}} \Big[ \underbrace{\mathbb{E}_{q_\phi(z|f_\theta(x^{(E)}))} \left[ -\log q_\psi(x|z) \right]}_{\mathcal{L}_{\text{LQ-recon}}} + \beta \underbrace{\text{KL}(q_\phi(z|f_\theta(x^{(E)})) \parallel r(z))}_{\mathcal{L}_{\text{HQ-prior}}} \Big]. \tag{14}$$

where the optimization is with respect to $\theta$.

### 4.2 TRAINING OBJECTIVE

The proposed loss $\mathcal{L}_{\text{IRIB-ELQ}}$ provides a simple method to solve image restoration tasks in extreme low-quality settings with minimal training. We now show how this loss is used as a practical training loss by examining each of its terms in detail under intuitions provided by previous work (Higgins et al., 2017; Wang et al., 2024b).

**LQ reconstruction term.** For a given random degradation $\mathcal{D}$, we set $q_\psi(x|z) = \mathcal{N}(\mathcal{D}(z), \sigma^2 I)$, which yields the LQ reconstruction loss as:

$$\mathcal{L}_{\text{LQ-recon}} = \mathbb{E}_{p(x)q_\phi(z|x)} \left[ \frac{1}{2\sigma^2} \parallel \mathcal{D}(z) - x \parallel_2^2 \right] + \text{const.} \tag{15}$$

The LQ reconstruction loss minimizes the $\ell_2$ distance of LQ image $x$ to the randomly degraded image $\mathcal{D}(z)$. However, a plain pixelwise $\ell_2$ objective disproportionately penalizes high-frequency discrepancies. This is ill-suited to our setting because $\mathcal{D}$ typically contains downsampling that discard

Figure 3: **Overall Pipeline.** Given an ELG image $x_{\text{ELQ}}$, the trainable projector $f_\theta$ produces $\hat{x}_{\text{LQ}}$; a frozen restoration model $g$ outputs $\hat{z}_{\text{HQ}}$. We simulate the *degrade-back* to obtain $\tilde{x}_{\text{LQ}} = \mathcal{D}(z_{\text{HQ}})$ and minimize a blur-aware LQ reconstruction loss $\mathcal{L}_{\text{LQ-recon}}$. To regularize toward the HQ distribution, we apply $\mathcal{L}_{\text{HQ-prior}}$ and $\mathcal{L}_{\text{HQ-fid}}$. Gradients for the terms only flow through $f_\theta$ while $g$ and the prior remain frozen. Recurisve prompt refinement can condition $f_\theta/g$ during inference time, enabling Look Forward Once refinement.

high-frequency details; as a result, multiple valid realizations of $\mathcal{D}(z)$ can differ in high-frequency content while being perceptually and semantically equivalent at LQ.

To improve robustness, we compare signals after a shared low-pass transform. Let $G_\tau$ denote a Gaussian blur operator with standard deviation $\tau$. We define the *blur-MSE* loss

$$\mathcal{L}_{\text{LQ-recon}} = \mathbb{E}_{p(x)\, q_\phi(z|x)} \left[ \frac{1}{2\sigma^2} \left\| G_\tau * \mathcal{D}(z) \;-\; G_\tau * x \right\|_2^2 \right], \tag{16}$$

where the same Gaussian kernel $G_\tau$ is applied to both $\mathcal{D}(z)$ and $x$. In the frequency domain, (16) weights discrepancies by down weighting high frequency mismatches that $\mathcal{D}$ cannot reliably preserve and attenuating noise-induced fluctuations.

In our practical implementation, the degradation $\mathcal{D}$ is set as that of the Real-ESRGAN pipeline (Wang et al., 2021). This degradation is shared among many state-of-the-art image restoration models; hence we can apply the trained $f_\theta$ to broaden the applicability of various different image restoration models. We examine the detailed effects in our experiments.

**HQ prior matching term.** We match $q_\phi(z|x)$ to an HQ prior $r(z) \approx p_{\text{HQ}}$ for the prior matching term $\mathcal{L}_{\text{HQ-prior}}$. Though this can be performed in many different ways (*e.g.*, adversarial loss, variational score distillation, f-divergence), for the context of image restoration we leverage the following prior matching loss originated from diffusion models:

$$\mathcal{L}_{\text{HQ-prior}} = \mathbb{E}_{x \sim p(x) q_\phi(z|x),\, \epsilon \sim \mathcal{N}(0,I)} \left[ \frac{1}{2} \left\| \hat{\epsilon}(z_t, t) \;-\; \hat{\epsilon}_{\text{HQ}}(z_t, t) \right\|_2^2 \right]. \tag{17}$$

Here, $\hat{\epsilon}$ is the student noise predictor and $\hat{\epsilon}_{\text{HQ}}$ is the (fixed) HQ prior's noise predictor, both evaluated on the same noised latent $z_t$. This term coincides with the variational score distillation (VSD) (Wang et al., 2024b) loss used in OSEDiff, and is optimized in parallel with the standard diffusion loss used to finetune the regularizer.

Beyond prior matching, we enforce sample-wise fidelity to the paired HQ target. We combine pixel, perceptual, and blur-aware terms:

$$\mathcal{L}_{\text{HQ-fid}} = \mathbb{E}_{(x,y),\, z \sim q_\phi(z|x)} \left[ \lambda_{\text{l2}} \left\| \hat{z} - z \right\|_2^2 + \lambda_{\text{LPIPS}} \, \text{LPIPS}(\hat{z}, z) + \lambda_{\text{blur}} \left\| G_k * \hat{z} - G_k * z \right\|_2^2 \right], \tag{18}$$

where $G_k$ is a Gaussian blur operator applied symmetrically to both images. In this case, the *blur-MSE* term acts as a robustness prior in the HQ domain. When the input is extremely degraded, multiple plausible restorations can share low-frequency structure while differing in fine details. Allowing this controlled freedom via $\lambda_{\text{blur}}$ and kernel size $k$ offers more realistic outputs with a small trade-off in pixel fidelity.

The total loss w.r.t. the network parameter $\theta$ becomes:

$$\mathcal{L}_{\text{total}} = \mathcal{L}_{\text{LQ-recon}} + \mathcal{L}_{\text{HQ-prior}} + \mathcal{L}_{\text{HQ-fid}}. \tag{19}$$

### 4.3 LOOK FORWARD ONCE

Decomposing the $\mathcal{X}_{\text{ELQ}} \rightarrow \mathcal{X}_{\text{HQ}}$ task to $\mathcal{X}_{\text{ELQ}} \rightarrow \mathcal{X}_{\text{LQ}} \rightarrow \mathcal{X}_{\text{HQ}}$ as in (13) also opens additional venues for refinement that we can exploit *during inference time* for improved restoration performance. In this work, we introduce Look Forward Once, a method for leveraging the LQ proxy $\hat{X}_{\text{LQ}}$ to perform auxiliary refinement via text prompt conditioning during inference time.

In our practical setup, the learnable projection model $f_\theta$ and the frozen IR model $g$ are both restoration models capable of referring to an input text condition. Let $(x_{\text{ELQ}}, \hat{x}_{\text{LQ}}, \hat{x}_{\text{HQ}})$ denote the given ELQ image, restored LQ image, and restored HQ image, respectively. Also, define $c_{\text{ELQ}}$ as the text condition used in $f_\theta$ for restoring $x_{\text{ELQ}}$ to $\hat{x}_{\text{LQ}}$, and $c_{\text{LQ}}$ as the text condition used in $g$ for restoring $\hat{x}_{\text{LQ}}$ to $\hat{x}_{\text{HQ}}$. The prompts $c_{\text{ELQ}}$ and $c_{\text{LQ}}$ are obtained with a prompt extraction module $Y$ as follows:

$$c_{\text{ELQ}} := Y(x_{\text{ELQ}}), \quad c_{\text{LQ}} := Y(\hat{x}_{\text{LQ}}) \tag{20}$$

In the proposed method, given the input $x_{\text{ELQ}}$ we *look forward once* during the restoration process to obtain an initial text condition $c_{\text{ELQ}}^{(1)}$ and its corresponding LQ image $\hat{x}_{\text{LQ}}^{(1)}$.

$$c_{\text{ELQ}}^{(1)} = Y(x_{\text{ELQ}}), \quad \hat{x}_{\text{LQ}}^{(1)} = f_\theta(x_{\text{ELQ}}; c_{\text{ELQ}}^{(1)}) \tag{21}$$

Then, we leverage $\hat{x}_{\text{LQ}}^{(1)}$ to create an improved text condition $c_{\text{ELQ}}^{(2)}$, which is subsequently used for the *second pass* of $\mathcal{X}_{\text{ELQ}} \rightarrow \mathcal{X}_{\text{LQ}}$. The corresponding $\hat{x}_{\text{LQ}}^{(2)}$ is used for the final $\mathcal{X}_{\text{LQ}} \rightarrow \mathcal{X}_{\text{HQ}}$ process to obtain $\hat{x}_{\text{HQ}}$. The final ELQ-to-HQ restoration process with LFO can be written as:

$$c_{\text{ELQ}}^{(2)} = Y(\hat{x}_{\text{LQ}}^{(1)}), \quad \hat{x}_{\text{HQ}} = g(f_\theta(x_{\text{ELQ}}; c_{\text{ELQ}}^{(2)})) \tag{22}$$

Here, the choice of prompt extraction module $Y$ is flexible ( *e.g.*, DAPE (Wu et al., 2024b) or VLM). By utilizing the intermediate successive prompts amidst the ELQ-to-HQ restoration process, we are able to provide better conditioning to be used for the ELQ-to-LQ restoration.

## 5 EXPERIMENTS

### 5.1 EXPERIMENTAL SETTINGS

We train at $512 \times 512$ resolution on the LSDIR dataset (Li et al., 2023). Both projection model and freezed IR model follow OSEDiff, with backbone Stable Diffusion v2.1 (Wu et al., 2024a). We extract image tags with RAM (Zhang et al., 2024b) as prompts; for unreliable tagging due to severe degradation, we apply prompt dropout by using a null prompt with probability $0.3$ for improved robustness. Synthetic degradations follow the Real-ESRGAN pipeline (Wang et al., 2021): our LQ setting uses the default configuration, while the extreme-LQ setting widens ranges for resizing factors, blur kernels (Gaussian $\sigma$ / generalized-Gaussian $\beta$), additive noise, and JPEG quality. We fine-tune only LoRA adapters of rank 4 with learning rate $5 \times 10^{-5}$ on $4 \times$ A100 GPUs. For testing, we evaluate our model on DIV2K and DIV8K train sets (800 and 1,500 images, respectively). All images are resized and center-cropped to $512 \times 512$ resolution. At test time, the same degradation settings as for training are applied and DAPE (Wu et al., 2024b) is used for prompt extraction.

### 5.2 COMPARISON RESULTS

We compare our IRIB setting against pretrained one-step IR models: real-ESRGAN, DiffBIR, SUPIR (Yu et al., 2024), and OSEDiff all collapse under extreme BIR settings. OSEDiff trained for the extreme ELQ→HQ degradation configuration is also compared. Our method employs LFO to iteratively extract prompts from intermediate LQ images to further refine the output.

**Qualitative Comparison.** Qualitative comparison is depicted in Figure 4. The naïve OSEDiff, trained solely on LQ images, struggles to recover severely degraded inputs and leaves noticeable artifacts. The fine-tuned OSEDiff improves visual quality but still exhibits irregular deformations and hallucinated textures. In contrast, combining an explicit projection onto the LQ manifold with LFO refinement restores ELQ inputs to semantically plausible, realistic images with higher fidelity.

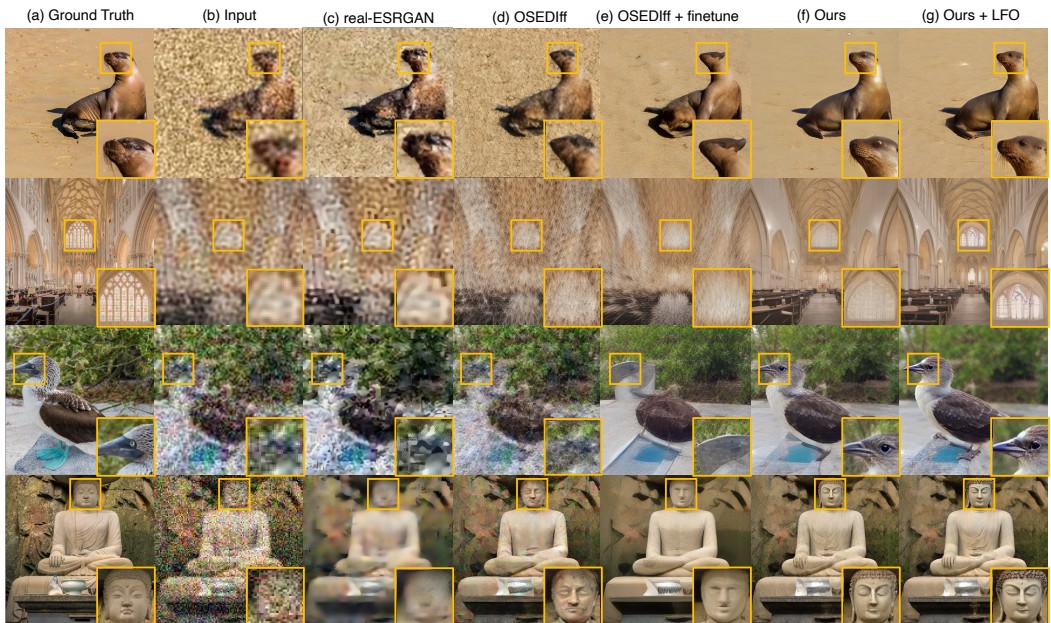

Figure 4: **Qualitative Comparison. (c-d)** One step restoration of OSEDiff and it's finetuned model; **(e-g)** Our methods and LFO prompt refinement variants. Although the fine-tuned OSEDiff improves image quality over the naïve baseline, it still produces occasional unrealistic artifacts. In contrast, the IRIB framework (ours) yields restorations that are both realistic and consistent with the ELQ input, effectively recovering the most probable image under severe degradations.

Table 1: Quantitative comparison. **Bold**: best, Underline: second-best.

| Method | Training Type | LFO | PSNR↑ | SSIM↑ | LPIPS↓ | DISTS↓ | FID↓ | NIQE↓ | MUSIQ↑ | CLIPIQA↑ |
|---|---|---|---|---|---|---|---|---|---|---|
| | | | | | | **DIV2K** | | | | |
| Real-ESRGAN | Pretrained | ✗ | 19.5666 | 0.4360 | 0.7020 | 0.3875 | 159.1040 | 8.6893 | 31.7782 | 0.3595 |
| DiffBIR | Pretrained | ✗ | 17.5312 | 0.3971 | 0.5738 | 0.3022 | 79.3112 | 5.9461 | 63.0555 | 0.6500 |
| SUPIR | Pretrained | ✗ | 18.3903 | 0.3218 | 0.6179 | 0.3487 | 110.0228 | 6.3272 | 61.5866 | 0.7158 |
| OSEDiff | Pretrained | ✗ | 19.6772 | 0.4397 | 0.4848 | 0.3029 | 81.5558 | 4.3314 | 63.8965 | 0.6202 |
| OSEDiff | ELQ→HQ | ✗ | **20.0958** | **0.4818** | 0.4275 | 0.2710 | 68.0802 | **4.1018** | 69.9307 | 0.6615 |
| OSEDiff + Ours | ELQ→LQ→HQ | ✗ | 20.0488 | 0.4797 | 0.4249 | 0.2621 | 62.9768 | 4.1814 | 70.3529 | 0.6829 |
| OSEDiff + Ours | ELQ→LQ→HQ | ×1 | 20.0084 | 0.4790 | **0.4261** | 0.2599 | 63.1425 | 4.1830 | 70.6405 | 0.6844 |
| OSEDiff + Ours | ELQ→LQ→HQ | ×2 | 20.0047 | 0.4790 | 0.4255 | **0.2594** | **62.9475** | 4.1875 | **70.6806** | **0.6848** |
| | | | | | | **DIV8K** | | | | |
| Real-ESRGAN | Pretrained | ✗ | 20.2318 | 0.4508 | 0.7096 | 0.3800 | 144.1936 | 8.6526; | 32.3691 | 0.3721 |
| DiffBIR | Pretrained | ✗ | 18.4392 | 0.4233 | 0.5512 | 0.2953 | 53.5166 | 6.0480 | 63.1401 | 0.6665 |
| SUPIR | Pretrained | ✗ | 19.2090 | 0.3545 | 0.6004 | 0.3331 | 84.2722 | 6.6578 | 60.8685 | 0.7096 |
| OSEDiff | Pretrained | ✗ | 20.5186 | 0.4651 | 0.4710 | 0.2944 | 58.1913 | 4.5441 | 63.9930 | 0.6309 |
| OSEDiff | ELQ→HQ | ✗ | **20.9333** | **0.5065** | 0.4082 | 0.2627 | 43.5674 | **4.2649** | 69.3240 | 0.6560 |
| OSEDiff + Ours | ELQ→LQ→HQ | ✗ | 20.8791 | 0.5040 | **0.4051** | 0.2535 | 40.7901 | 4.3803 | 69.7483 | 0.6803 |
| OSEDiff + Ours | ELQ→LQ→HQ | ×1 | 20.8489 | 0.5038 | 0.4057 | 0.2516 | 40.5819 | 4.3829 | 69.8903 | 0.6801 |
| OSEDiff + Ours | ELQ→LQ→HQ | ×2 | 20.8421 | 0.5039 | 0.4053 | **0.2505** | **40.5523** | 4.3719 | **70.0020** | **0.6806** |

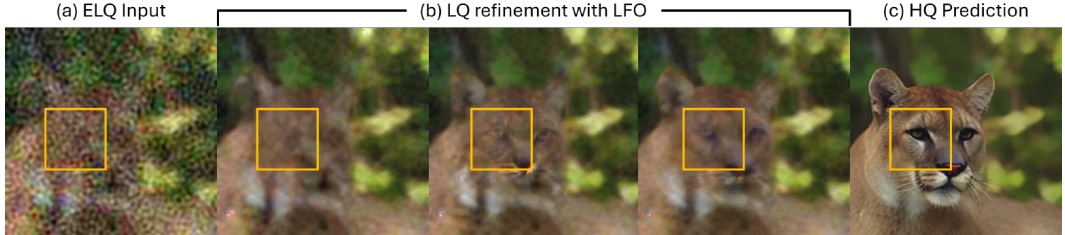

Figure 5: **LQ refinement with LFO. (a)** The ELQ input. **(b)** From left to right: the initial LQ sample; the refined LQ sample after 1× LFO; the refined LQ sample after 2× LFO. Iterative refinement of the LQ sample with LFO forms coarse structure (*e.g.*, eye) in LQ. **(c)** The final HQ output.

Table 2: Plug-and-Play results of our projection module $f_\theta$ on existing BIR models. **Bold**: best.

| | DIV2K | | | | | | | |
| Method | PSNR↑ | SSIM↑ | LPIPS↓ | DISTS↓ | FID↓ | NIQE↓ | MUSIQ↑ | CLIPIQA↑ |
|---|---|---|---|---|---|---|---|---|
| S3Diff | 19.5941 | 0.4379 | 0.4583 | 0.2830 | 74.7576 | 4.8987 | 63.5064 | 0.6411 |
| S3Diff + Ours | **19.8582** | **0.4581** | **0.4075** | **0.2382** | **58.3893** | **4.3436** | 70.8872 | **0.6975** |
| SeeSR | 19.1646 | 0.4430 | 0.4738 | 0.2507 | 69.3567 | 4.4721 | 71.1555 | **0.7424** |
| SeeSR + Ours | **19.9312** | **0.4632** | **0.4506** | **0.2374** | **61.2615** | **4.1355** | **71.5719** | 0.7320 |

| | DIV8K | | | | | | | |
| Method | PSNR↑ | SSIM↑ | LPIPS↓ | DISTS↓ | FID↓ | NIQE↓ | MUSIQ↑ | CLIPIQA↑ |
|---|---|---|---|---|---|---|---|---|
| S3Diff | 20.4950 | 0.4630 | 0.4490 | 0.2754 | 53.0353 | 4.9418 | 62.9439 | 0.6386 |
| S3Diff + Ours | **20.7011** | **0.4813** | **0.3880** | **0.2334** | **36.7050** | **4.5596** | 70.5182 | **0.6935** |
| SeeSR | 20.0719 | 0.4693 | 0.4622 | 0.2476 | 43.4365 | 4.6056 | 70.3421 | **0.7395** |
| SeeSR + Ours | **20.8649** | **0.4921** | **0.4345** | **0.2345** | 39.6162 | **4.3091** | **70.7543** | 0.7285 |

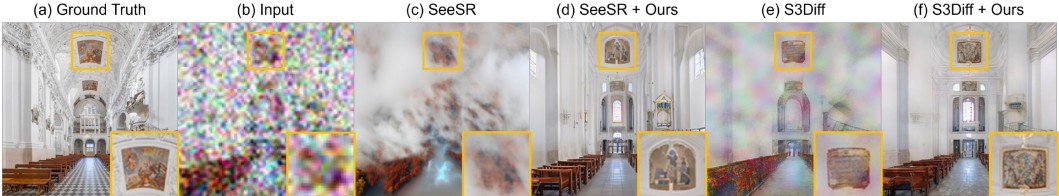

(a) Ground Truth    (b) Input    (c) SeeSR    (d) SeeSR + Ours    (e) S3Diff    (f) S3Diff + Ours

Figure 6: **Plug-and-Play with Existing BIR Models. (c,e)** Direct ELQ restoration of baselines; **(d,f)** LQ restoration of baselines after IRIB projection. Integrating our method enables SeeSR and S3Diff to produce high-fidelity reconstructions from ELQ inputs with little to no hallucination.

**Quantitative Comparison.** Quantitative results are given in Table 1, and ablation study provided in Appendix A. We use PSNR, SSIM, LPIPS (Zhang et al., 2018), DISTS for fidelity metrics and NIQE (Zhang et al., 2015), MUSIQ (Ke et al., 2021), CLIPIQA (Wang et al., 2023) for perceptual metrics. To evaluate the realism of the restored images we use Frechet Inception Distance (FID). Given that severely degraded inputs admit multiple plausible restorations, we prioritize perceptual realism while preserving global structure. Our method produces semantically faithful, visually plausible results with only minor trade-offs in pixel-level fidelity (*e.g.*, PSNR/SSIM). In particular, recursive LFO refinement consistently improves perceptual metrics yielding higher perceived quality.

**Training Efficiency Comparison.** Training graphs for baseline ELQ→HQ training and our method of ELQ→LQ→HQ training is compared in Figure 7. For both cases, the number of training parameters are kept the same, as we fine-tune LoRA adapters of the same rank (*i.e.*, 4). Our method achieves superior PSNR and FID with a much smaller number of training iterations, showing that our decomposition benefits both speed and performance. Note that increasing the LoRA rank for the baseline *does not* improve performance for the ELQ→HQ baseline, as explained in Appendix B.

### 5.3 PLUG-AND-PLAY WITH EXISTING BASED BIR MODELS

We integrate our ELQ → LQ projection with existing IR models SeeSR (Wu et al., 2024b) and S3Diff (Zhang et al., 2024a), trained on the same Real-ESRGAN degradation pipeline as our setup. Figure 6 contrasts two pipelines: (i) Directly feeding the ELQ input to the IR model, and (ii) projecting ELQ to the LQ manifold using our method, then applying the IR model for LQ→HQ restoration. SeeSR and S3Diff is kept frozen and evaluated with the official inference settings. Directly using an IR model for ELQ images exhibits softened details and deformations. In contrast, our method reduces artifacts and recovers sharper, more coherent structures. Quantitatively (Table 2), our method yields substantial improvement across pixel fidelity, perceptual quality, and realism metrics.

### 5.4 ABLATION STUDY

**Individual components of the $\mathcal{L}_{\text{IRIB-ELQ}}$ loss.** Thorough analysis of each individual component in Eq. 19 is provided in Table 3. The ablation results give sufficient insight into the purposes of each individual component. When either the $\mathcal{L}_{\text{LQ-recon}}$ or $\mathcal{L}_{\text{HQ-fid}}$ component is ablated, there is strict drop in reference-based consistency metrics (*i.e.*, PSNR, SSIM, LPIPS, DISTS). Thus, these loss terms enable the model to generate images consistent with the ELQ input. When the $\mathcal{L}_{\text{HQ-prior}}$ component is

Table 3: Ablation study of the $\mathcal{L}_{\text{IRIB-ELQ}}$ loss. **Bold**: best, Underline: second-best.

| $\mathcal{L}_{\text{LQ-recon}}$ | $\mathcal{L}_{\text{HQ-prior}}$ | $\mathcal{L}_{\text{HQ-fid}}$ | PSNR↑ | SSIM↑ | LPIPS↓ | DISTS↓ | FID↓ | NIQE↓ | MUSIQ↑ | CLIPIQA↑ |
|---|---|---|---|---|---|---|---|---|---|---|
| | | | **DIV2K** | | | | | | | |
| ✗ | ✓ | ✓ | 19.2451 | 0.4592 | 0.4818 | 0.2801 | 71.5057 | 4.5954 | **72.7040** | 0.7303 |
| ✓ | ✗ | ✓ | **20.0718** | **0.4898** | 0.4464 | 0.2808 | 75.1813 | 5.4442 | 70.3964 | **0.7449** |
| ✓ | ✓ | ✗ | 16.7439 | 0.3764 | 0.6740 | 0.3317 | 102.6394 | 4.3963 | 71.3685 | 0.6942 |
| ✓ | ✓ | ✓ | 20.0488 | 0.4797 | **0.4249** | **0.2621** | **62.9768** | **4.1814** | 70.3529 | 0.6829 |
| | | | **DIV8K** | | | | | | | |
| $\mathcal{L}_{\text{LQ-recon}}$ | $\mathcal{L}_{\text{HQ-prior}}$ | $\mathcal{L}_{\text{HQ-fid}}$ | PSNR↑ | SSIM↑ | LPIPS↓ | DISTS↓ | FID↓ | NIQE↓ | MUSIQ↑ | CLIPIQA↑ |
| ✗ | ✓ | ✓ | 20.0605 | 0.4874 | 0.4583 | 0.2719 | 47.5777 | 4.7432 | **72.0902** | 0.7238 |
| ✓ | ✗ | ✓ | **20.9121** | **0.5133** | 0.4290 | 0.2699 | 48.2483 | 5.4185 | 69.7655 | **0.7401** |
| ✓ | ✓ | ✗ | 17.5600 | 0.4126 | 0.6497 | 0.3230 | 77.3818 | 4.5927 | 70.2459 | 0.6955 |
| ✓ | ✓ | ✓ | 20.8791 | 0.5040 | **0.4051** | **0.2535** | **40.7901** | **4.3803** | 69.7483 | 0.6803 |

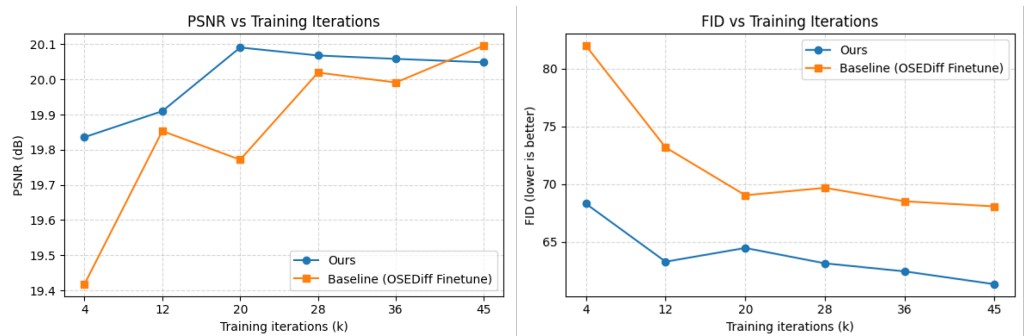

Figure 7: **Training graphs for baseline ELQ→HQ training and our method of ELQ→LQ→HQ training.** Our method quickly achieves superior performance with less training iterations.

ablated, we observe degraded performance on metrics that evaluate perceptual quality (*e.g.*, LPIPS, DISTS, FID, NIQE). Overall, our final objective provides an adequate balance between fidelity and perceptual quality.

**Ablation study of $\lambda_{\text{blur}}$.** Ablation study regarding the hyperparameter $\lambda_{\text{blur}}$ of Eq. (18) is provided in Section A of the Appendix. Results show that users are able to balance fidelity and perceptual quality *as preferred* by adjusting $\lambda_{\text{blur}}$.

## 6 CONCLUSION

Extreme Blind Image Restoration (EBIR) remains under-explored. We introduce an EBIR framework cast in the Information Bottleneck (IB) paradigm, deriving the *Image Restoration Information Bottleneck* (IRIB) objective that couples a blur-aware LQ reconstruction term with an HQ prior-matching term, optionally complemented by sample-wise HQ fidelity losses. By factorizing restoration into an ELQ→LQ→HQ pipeline and training only a projector $f_\theta$ while reusing a frozen backbone $g$, we shrink the search space, stabilize optimization under compounded degradations, and expose an intermediate LQ representation that enables *Look Forward Once* (LFO) prompt refinement. Our framework is modular thus providing broad applicability; existing BIR models (*i.e.*, SeeSR, S3Diff) can be improved in a plug-and-play setting. Future work includes learning degradations beyond Real-ESRGAN and extending to additional modalities to enhance generalization.

**Limitations.** One limitation of our framework is that the pretrained IR model $g$ used for training the $f_\theta$ projection must be a single step model for practical usage, since gradients would have to flow through the model multiple times for a multi-step model (*e.g.*, diffusion-based). Furthermore, the prompt extraction module $Y$ is not able to produce optimal results for input ELQ cases. Thus, future work could perform additional fine-tuning of $Y$ or leverage VLMs to achieve higher performance.

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

# A   ABLATION STUDY

**$\lambda_{\text{blur}}$ ablation study**   We augment the HQ fidelity objective as in 18 with a blur–MSE term to reflect the ambiguity under extreme degradations: multiple plausible restorations can share similar low-frequency structure while differing in fine details. Concretely, we consider

$$\mathcal{L}_{\text{HQ}}^{\text{blur}} \;=\; \lambda_{\text{blur}} \, \| \, G_k * \hat{z} \; - \; G_k * z \, \|_2^2 \tag{23}$$

where $G_\sigma$ is a Gaussian low-pass filter, and ablate $\lambda_{\text{blur}} \in \{0, 0.5, 1, 2\}$ while keeping the $\lambda_{\text{LPIPS}}$ and $\lambda_{\text{l2}}$ fixed to 1. The PSNR–MUSIQ and PSNR–FID plots in Figure 8 visualize the trade-off: as $\lambda_{\text{blur}}$ increases, models move toward higher pixel-level fidelity (PSNR↑) at the expense of perceptual realism and distributional quality (MUSIQ ↓, FID ↑). Thus, *users are able to freely balance fidelity and perceptual quality as preferred by adjusting $\lambda_{blur}$.* Increasing $\lambda_{blur}$ yields higher fidelity than fine-tuned baselines, but at the cost of perceptual quality. Such effect is monotonic across both DIV2K and DIV8K datasets, with the fine-tuned baseline (*i.e.*, OSEDiff specifically fine-tuned for ELQ→LQ mapping) providing an operational reference on these planes. For fair comparison with the baseline we do not apply LFO in our ablations; iteratively applying LFO during inference time further increases performance of our method.

Table 4: Quantitative comparison of varying $\lambda_{\text{blur}}$ compared to a fine-tuned baseline. **Bold**: best.

| | **DIV2K** | | | | | | | |
| Method | PSNR↑ | SSIM↑ | LPIPS↓ | DISTS↓ | FID↓ | NIQE↓ | MUSIQ↑ | CLIPIQA↑ |
|---|---|---|---|---|---|---|---|---|
| OSEDiff Fine-tuned | 20.0958 | 0.4818 | 0.4275 | 0.2710 | 68.0802 | 4.1018 | 69.9307 | 0.6615 |
| OSEDiff + Ours ($\lambda_{\text{blur}} = 0$) | 19.5735 | 0.4300 | 0.4725 | 0.3003 | **62.4275** | **3.3994** | **71.1609** | **0.7004** |
| OSEDiff + Ours ($\lambda_{\text{blur}} = 0.5$) | 20.0488 | 0.4797 | 0.4249 | **0.2621** | 62.9768 | 4.1814 | 70.3529 | 0.6829 |
| OSEDiff + Ours ($\lambda_{\text{blur}} = 1$) | 20.1332 | 0.4838 | **0.4251** | 0.2622 | 63.4009 | 4.3263 | 69.5339 | 0.6753 |
| OSEDiff + Ours ($\lambda_{\text{blur}} = 2$) | **20.2779** | **0.4871** | 0.4263 | 0.2644 | 63.3627 | 4.4081 | 68.9951 | 0.6774 |
| | **DIV8K** | | | | | | | |
| Method | PSNR↑ | SSIM↑ | LPIPS↓ | DISTS↓ | FID↓ | NIQE↓ | MUSIQ↑ | CLIPIQA↑ |
| OSEDiff Fine-tuned | 20.9333 | 0.5065 | 0.4082 | 0.2627 | 43.5674 | **4.2649** | 69.3240 | 0.6560 |
| OSEDiff + Ours ($\lambda_{\text{blur}} = 0$) | 20.7544 | 0.5016 | **0.4044** | **0.2510** | **40.2708** | 4.2857 | 70.1484 | **0.6830** |
| OSEDiff + Ours ($\lambda_{\text{blur}} = 0.5$) | 20.8791 | 0.5040 | 0.4051 | 0.2535 | 40.7899 | 4.3803 | 69.7483 | 0.6803 |
| OSEDiff + Ours ($\lambda_{\text{blur}} = 1$) | 20.9850 | 0.5071 | 0.4076 | 0.2554 | 41.2787 | 4.4901 | 69.1095 | 0.6760 |
| OSEDiff + Ours ($\lambda_{\text{blur}} = 2$) | **21.1423** | **0.5108** | 0.4075 | 0.2563 | 41.4256 | 4.5464 | 68.5735 | 0.6783 |

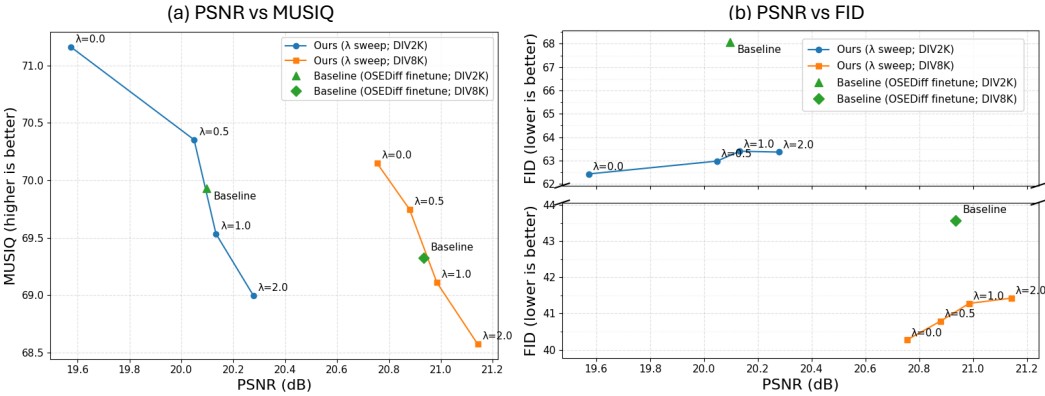

Figure 8: **Trade-off induced by $\lambda_{\text{blur}}$.** (a,b) PSNR-MUSIQ, PSNR-FID curve. Perceptual quality decreases as $\lambda_{\text{blur}}$ increases.

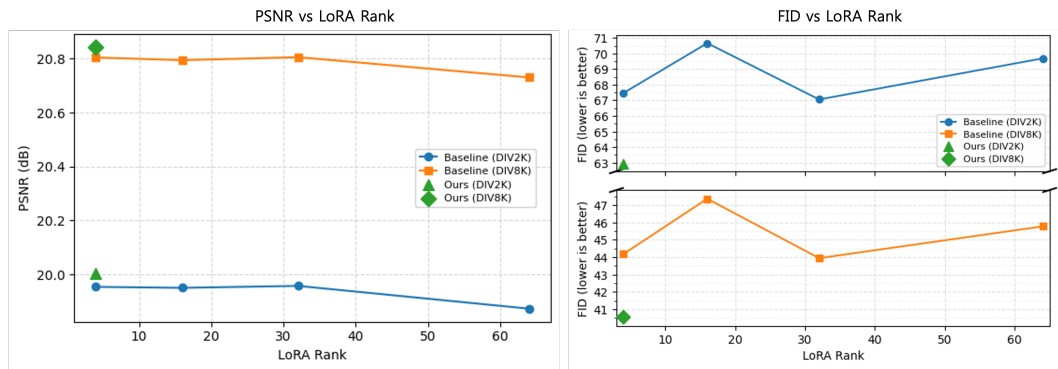

Figure 9: **Comparison between different LoRA ranks for the baseline ELQ→HQ training method.** Even if the number of LoRA ranks increases, baseline performance does not increase; rather, a sharp *decrease* in PSNR can be observed for high LoRA ranks (*e.g.*, 64). Our method of ELQ→LQ→HQ training shows better performance than the baseline regardless of LoRA rank.

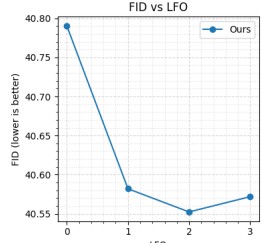

Figure 10: FID values per LFO iteration.

Table 5: Computational complexity of different methods.

| Model | Inference Time (ms) | MACs (G) | # Params (Total/Trainable) |
|---|---|---|---|
| Baseline | 0.1562 | 2265 | 1775M / 8.5M |
| Ours (no LFO) | 0.3067 | 4530 | 3550M / 8.5M |
| Ours (LFO ×1) | 0.4593 | 6795 | 3550M / 8.5M |
| Ours (LFO ×2) | 0.6187 | 9060 | 3550M / 8.5M |
| Ours (LFO ×3) | 0.7575 | 11325 | 3550M / 8.5M |

## B  LIMITATION OF ELQ→HQ TRAINING

We provide thorough comparison between different LoRA ranks of 4, 16, 32, 64 for the baseline ELQ→HQ training in Figure 9. Increasing the LoRA rank in this way gives insight into the correlation between the number of trainable parameters and model performance. We find that even if the LoRA rank increases, baseline performance does not increase; rather, a sharp decrease in PSNR is observed for the high LoRA rank of 64. Compared to baselines of all LoRA ranks, our method of ELQ→LQ→HQ training shows consistently better performance even with LoRA rank of 4.

## C  COMPUTATIONAL COMPLEXITY

Details regarding computational complexity during inference time are summarized in Table 5. Furthermore, we provide additional analysis of how FID saturates over LFO iterations in Figure 10. We empirically found that two LFO iterations are optimal in terms of the trade-off between computational cost and performance gain, where FID reaches its best value at the second iteration and then oscillates. This can be interpreted as suggesting that the maximum number of iterations from which we can extract meaningful text prompts is around 2 to 3.

## D  TEXT PROMPT EXAMPLES FOR LFO

Text prompt examples with and without LFO during the restoration process are given in Figure 11. Without LFO, the caption model often fails to output meaningful prompts from ELQ images (*e.g.*, a single prompt "floor" to explain a complex image). As LFO progresses, the caption model is able to generate more meaningful prompts to guide the restoration process (*e.g.*, an informative prompt "dog, gray, neckband, sit, stand").

# E    REAL-WORLD EXAMPLES

Real-world examples of how our restoration method can be used is provided in Figure 12. Historical and archival photos often suffer from heavy degradation due to aging and poor storage. Such heavily degraded images (obtained from (Singh, 2021)) can be restored to high quality with our method.

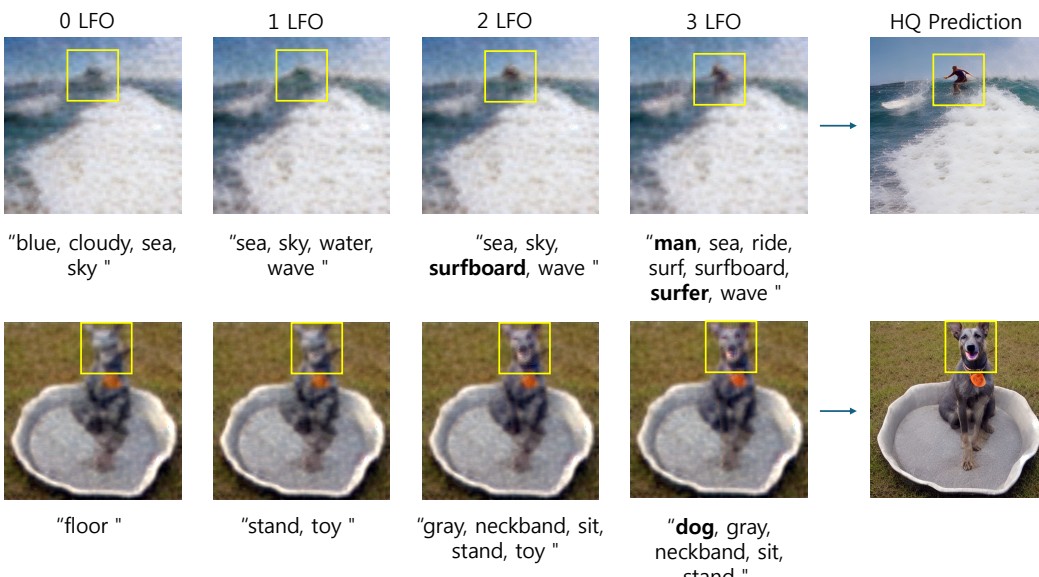

Figure 11: **Change of Text Prompt with LFO Iteration.** Without using LFO, the caption model often fails to output meaningful prompts from heavily degraded ELQ images. For example, a single prompt "floor" is used to explain a complex image. As LFO is utilized iteratively, the caption model is able to generate more meaningful prompts. The final prompt "dog, gray, neckband, sit, stand" contains sufficient information to effectively guide the restoration process.

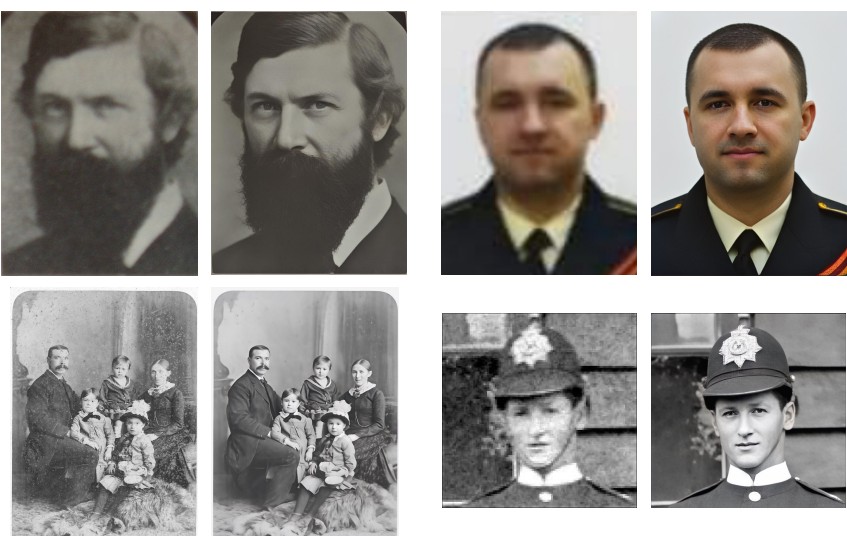

Figure 12: **Restoration of old photos with our method.** Highly degraded old photos can be effectively restored with our method.

# F ADDITIONAL QUALITATIVE COMPARISON RESULTS

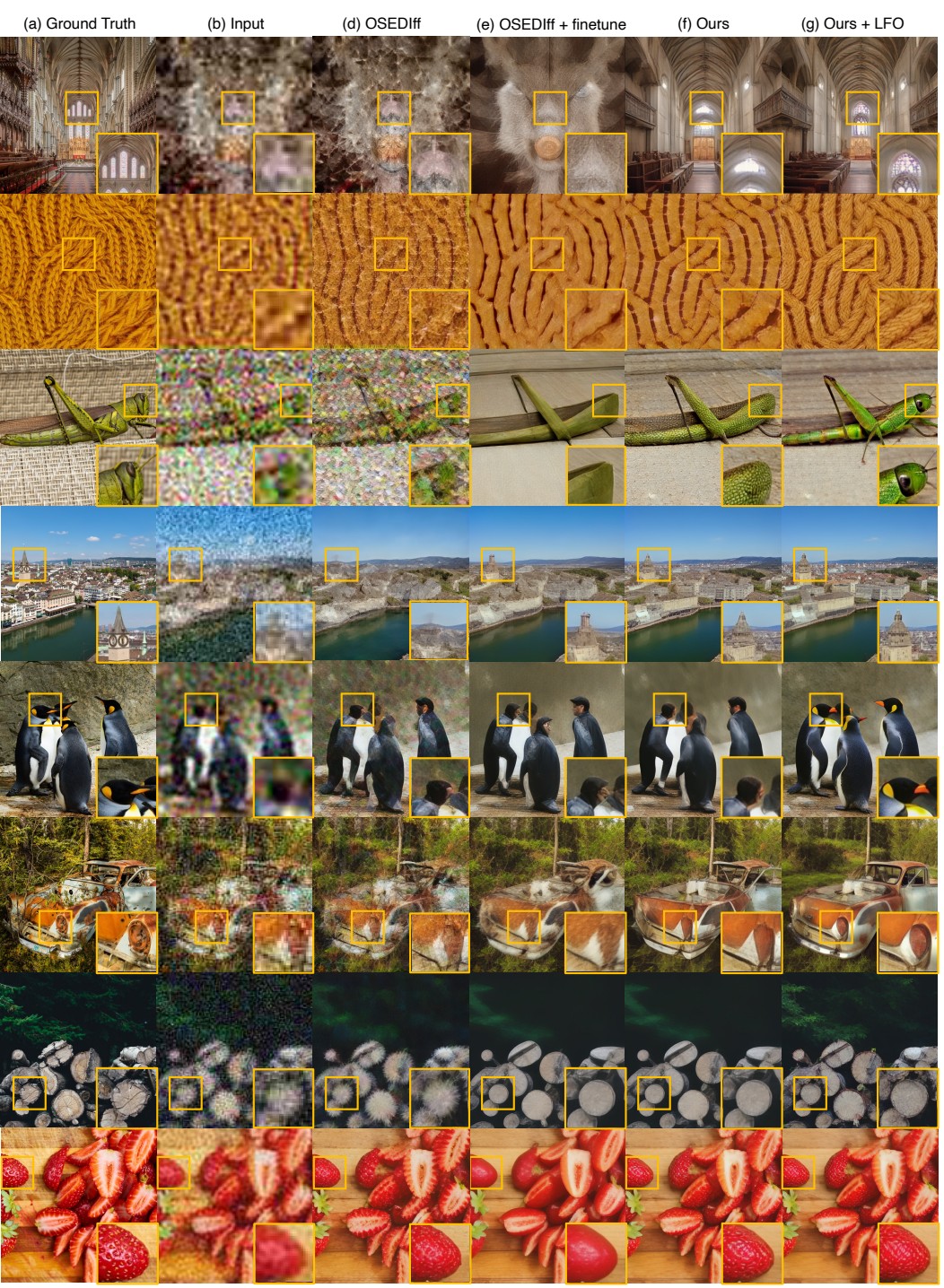

Figure 13: **Qualitative Comparison Results.**

# G ADDITIONAL QUALITATIVE RESULTS OF LQ REFINEMENT WITH RECURSIVE LFO

(a) ELQ Input          (b) LQ refinement with LFO          (c) HQ Prediction

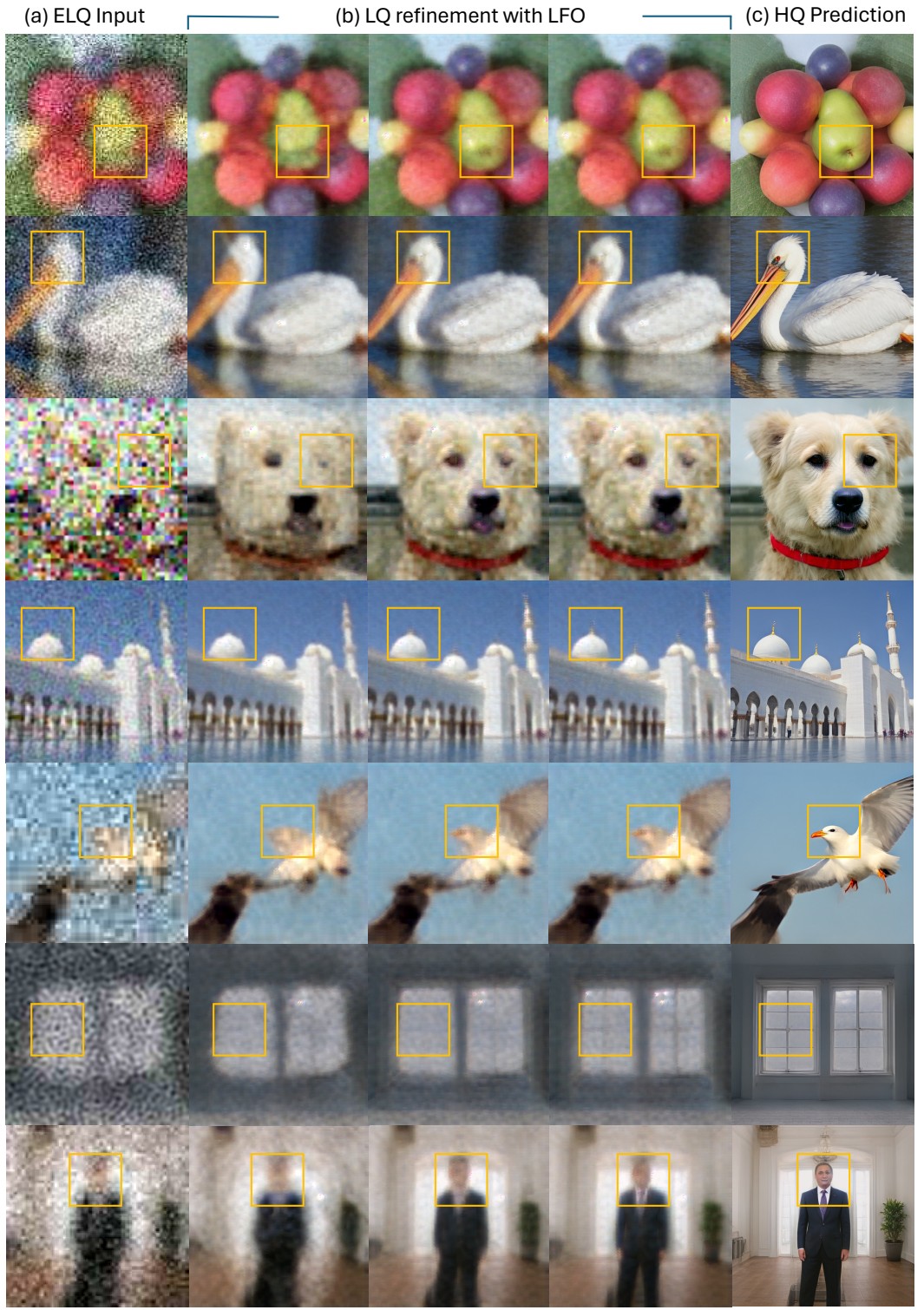

Figure 14: **Qualitative Results of LQ Refinement with Recursive LFO.**

## H    ADDITIONAL QUALITATIVE RESULTS OF PLUG-AND-PLAY WITH SEESR

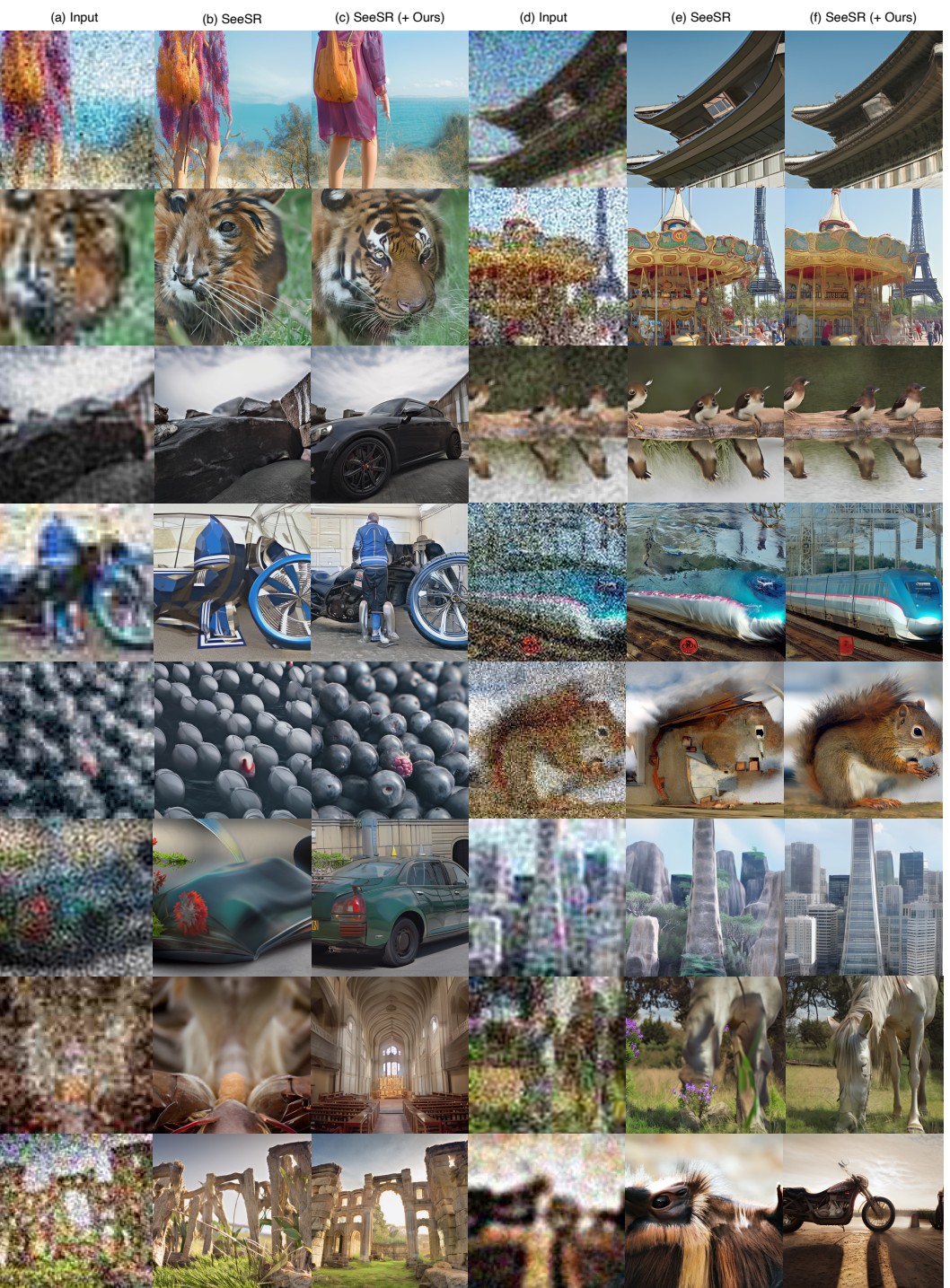

Figure 15: **Qualitative Results of Plug-and-Play with SeeSR.**

# I  ADDITIONAL QUALITATIVE RESULTS OF PLUG-AND-PLAY WITH S3DIFF

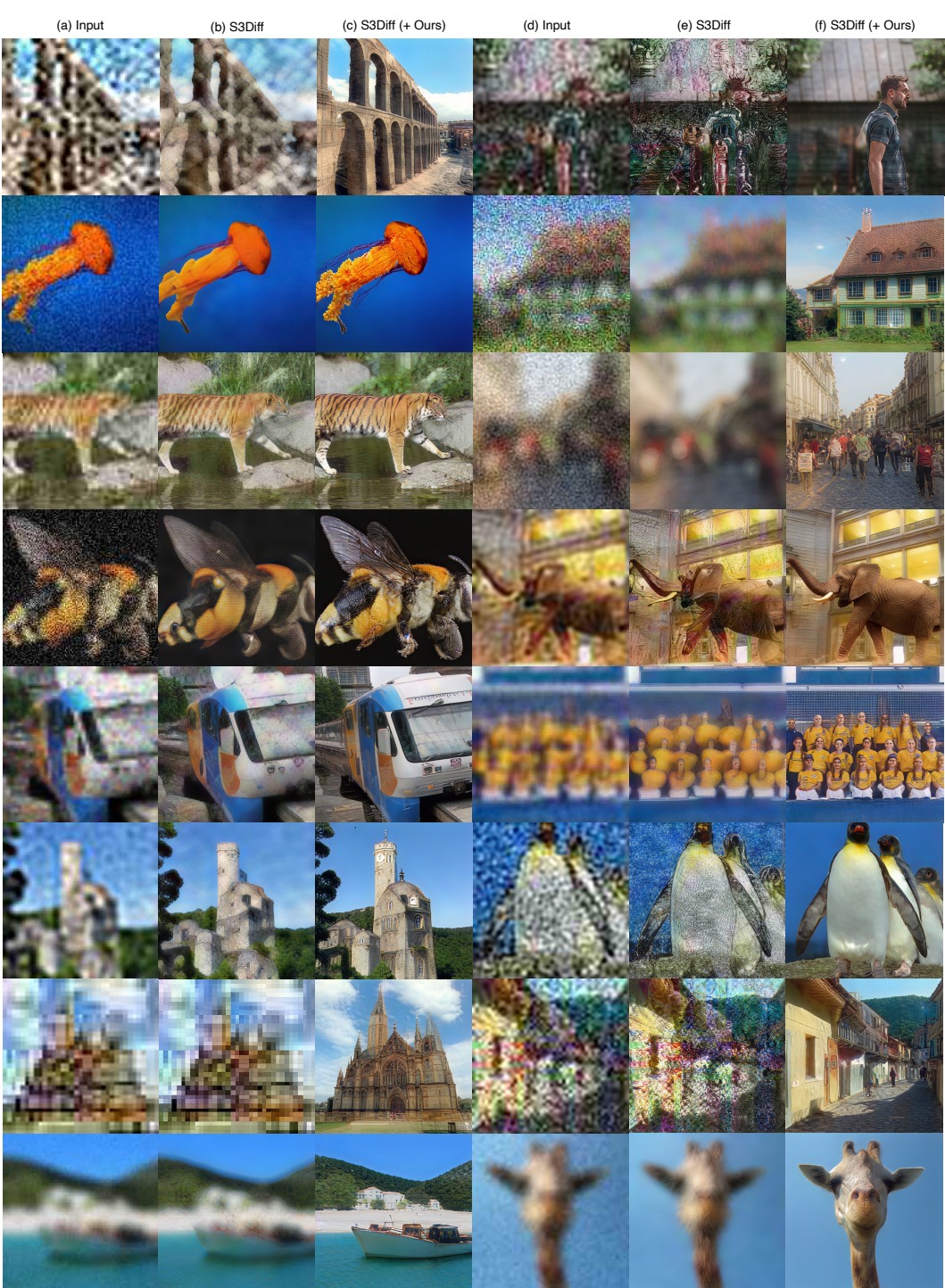

Figure 16: **Qualitative Results of Plug-and-Play with S3Diff.**

## J  THE USE OF LARGE LANGUAGE MODELS (LLMs)

LLMs were not involved in research ideation or methodological design and were only used for the purpose of minor expression refinement. The authors retain full responsibility for all scientific content.

