# OpenReview forum: "Extreme Blind Image Restoration via Prompt-Conditioned Information Bottleneck"
_ICLR.cc/2026/Conference — Submitted to ICLR 2026_

### Official Review · Reviewer_J7vm · 2025-10-27

**Soundness:** 3
**Presentation:** 3
**Contribution:** 2
**Rating:** 6
**Confidence:** 4

**Summary:**

This paper proposes a progressive method for extreme blind image restoration. It leverages LQ images as a proxy between ELQ images and HQ images, and devise a cycle consistency constraints between proxy LQ and synthesized LQ through information bottleneck theory. Additionally, the authors develop a prompt refinement strategy (LFO) based on the intermediate LQ results, further improving the restoration quality of HQ. The experiments demonstrate impressive visual results for extreme blind images.

**Strengths:**

1. The problem definition is quite interesting and rarely studied. The restoration results of ELQ -> LQ -> HQ images are obviously much better than ELQ -> HQ finetuned results.
2. The blur-MSE loss is a thoughtful and reasonable design for cycle constraining predicted LQ and synthesized LQ. The HQ-prior and HQ-fid loss functions are also reasonable.
3. The visual results of the restored ELQ images are impressive.
4. Experimental results in Table 2 demonstrates good generalization abilities of the proposed method.

**Weaknesses:**

1. The core design of this paper lies in the progressive restoration process and the cycle consistency constraint, both of which are well-studied in image restoration. Hence, the contribution of this paper feels somewhat incremental.
2. I think some real-world extremely distorted samples are needed for validation. For some distorted images in Figures 6 and 8, the inputs are severely degraded with almost no visible high-frequency details to help infer the content. I am also curious about what the text prompts are and how they contribute to the restoration process. Can the caption model output meaningful prompts from these ELQ images?
3. It seems a mistake in Eq. 21: g is not needed to generate x_LQ.

**Questions:**

1. I wonder about the practicality of this method. In what real-world scenarios would images be so severely degraded that they need such restoration?

---

> ### Author Response · Authors · 2025-11-24
>
> **W1. The progressive restoration process and the cycle consistency constraint are both well-studied in image restoration, making the contribution of this paper feel somewhat incremental.**
>
> **A.** Though progressive restoration and cycle consistency are well-studied approaches, our novelty lies in *what* main problem we solve and *how* progressive restoration and cycle consistency can be applied to solve it under innovative theoretical interpretations.
>
> In our work, the main problem is to solve *extreme* BIR, which we define as restoring images degraded *beyond* the training settings of conventional models. This is a rarely studied problem, as the majority of work focuses more on better restoration *within* conventional training settings (e.g., the Real-ESRGAN degradation pipeline). We bring a novel application of the Information Bottleneck (IB) theory to solve this under-explored problem, and propose a carefully designed loss under this theoretical interpretation.
>
> Thus, existing methods apply progressive restoration and cycle consistency to derive better approximations for the LQ→HQ mapping. In contrast, we introduce a new set of *extremely* low quality (ELQ) images and show how the highly intractable ELQ→HQ mapping can be effectively solved by leveraging the IB theory. As we unravel the IB theory for extreme BIR, progressive restoration and cycle consistency are introduced as technical details that emerge naturally from our theoretical interpretations.
>
> The theoretically-grounded general framework we construct opens up new possibilities and avenues for research. The LFO strategy we introduce is one such example of how our newly built framework allows for previously unattainable prompt refinement during inference time. Plug-and-play utilization of our projection module is another example: pre-trained BIR models can be easily enhanced to the domain of extreme BIR by simply plugging in a pre-trained module. In summary, our novelties lie in the innovative IRIB framework we propose, and the auxiliary refinement strategies that can be derived under this new framework.
>
> ---
> **W2, Q1. Some real-world extremely distorted samples are needed for validation. Also, in what real-world scenarios would images be so severely degraded that they need such restoration?**
>
> **A.** Thanks for the suggestion. While the paper focuses on extremely degraded images, such conditions are in fact common in several real-world scenarios where re-acquisition is impossible. For example, **historical and archival photos** often suffer from heavy degradation due to aging and poor storage, and **low-bitrate surveillance footage** requires visibility enhancement for adequate usage. We have added examples of highly degraded real-world old photo samples and their restoration results in **Appendix E** of our updated manuscript.
>
> ---
> **W2. What are the text prompts and how do they contribute to the restoration process? Can the caption model output meaningful prompts from these ELQ images?**
>
> **A.** Text prompt examples with and without LFO during the restoration process are given in **Appendix D** of our updated manuscript. Without LFO, the caption model often fails to output meaningful prompts from ELQ images (e.g., a single prompt "floor" to explain a complex image). As LFO progresses, the caption model is able to generate more meaningful prompts to guide the restoration process (e.g., an informative prompt "dog, gray, neckband, sit, stand").
>
> ---
> **W3. There is a mistake in Eq. 21: g is not needed to generate x_LQ.**
>
> **A.** Thank you for pointing this out. We have fixed the equation in our updated manuscript.

---

> > ### Comment · Reviewer_J7vm · 2025-11-26
> >
> > Thanks for the authors' response. Overall my concerns regarding text prompt and real-world practicability are addressed by newly added examples in Appendix. I partially agree with the authors’ statement regarding the novelty. However, I believe the main strength of this work lies in its interesting problem definition and its practical usefulness. I am inclined to accept this paper, and I would like to raise my score to 7 once that option becomes available.

---

> > > ### Author Response · Authors · 2025-11-26
> > > **Appreciation of Inclination to Accept**
> > >
> > > We thank the reviewer for their inclination to accept and are glad that all remaining concerns regarding text prompts and real-world practicality are addressed.

---

### Official Review · Reviewer_augB · 2025-10-31

**Soundness:** 2
**Presentation:** 2
**Contribution:** 2
**Rating:** 4
**Confidence:** 4

**Summary:**

This paper addresses the problem of Extreme Blind Image Restoration (EBIR), where images suffer from severe, compounded degradations beyond the scope of typical Blind Image Restoration (BIR) models. The authors propose a framework that decomposes this challenging task into two simpler steps: 1) a trainable "projector" ($f_{\theta}$) maps the extreme low-quality (ELQ) input onto an intermediate, less-degraded (LQ) manifold , and 2) a frozen, off-the-shelf BIR model ($g$) restores the intermediate LQ image to high-quality (HQ). The core of the method lies in the training of this projector, for which the authors derive a theoretically-driven objective from the Information Bottleneck (IB) principle.

**Strengths:**

The paper is generally well-written and clearly structured. The core methodology is well-illustrated in Figure 2 and Figure 3, which aids in understanding the proposed pipeline

**Weaknesses:**

1.Novelty of the Core Idea: The claim of novelty regarding the use of the Information Bottleneck (IB) principle for image restoration needs to be more carefully justified and contextualized. The application of IB to restoration and super-resolution is an active area of research. For instance, (Hsu et al., 2024) and (Zhu et al., 2024) have already explored IB-based objectives for super-resolution tasks. More directly, (Gao et al., 2025) also proposes an IB-based framework (InfoBFR) as a plug-and-play module for blind face restoration. The authors must provide a clearer differentiation of their specific IRIB formulation and its application as a pre-processor for general-domain EBIR against this existing literature to solidify the paper's novel contributions.

2.Insufficient Experimental Validation: The paper's central premise is that a direct, end-to-end (E2E) mapping from ELQ to HQ is "challenging" or "intractable,"  thus justifying the proposed two-stage decomposition. However, the experimental evidence provided is insufficient to support this strong claim. The primary baseline for the E2E approach appears to be "OSEDiff + finetune". As the paper notes, this finetuning was done using LoRA adapters. This is not equivalent to a full, rigorous training of the model from scratch on the ELQ $\rightarrow$ HQ task. This baseline is weak and does not represent the true capability of a fully trained diffusion model.This premise seems to challenge the consensus in the field. Foundation diffusion models are fundamentally designed to restore highly structured images from pure Gaussian noise—a task that arguably involves a more "massive domain gap" than restoring from ELQ inputs, which still retain low-frequency structural information.Furthermore, extensive work in extreme-low-bitrate image compression, such as (Relic et al., 2024) and (Xu et al., 2025), has clearly demonstrated that diffusion-based models can be effectively trained to restore high-fidelity images from severely distorted and information-poor inputs.Given this, the claim that the direct ELQ $\rightarrow$ HQ mapping is "intractable" is unconvincing. The experiments do not provide a fair comparison and are insufficient to prove the superiority of the proposed two-stage approach over a properly and fully trained E2E baseline.


3.Ambiguous Motivation and Contribution: The paper's motivation is unclear. It seems to oscillate between two different claims:Fundamental Claim: The ELQ $\rightarrow$ HQ task is fundamentally "ill-posed" and "intractable,"  requiring the proposed decomposition. Practical Claim: The solution (a pre-processor for a frozen backbone ) suggests the motivation is practical—avoiding the high computational cost of retraining large diffusion models.If the motivation is practical (which is a valid contribution), the experimental setup should compare against other parameter-efficient adaptation techniques. For example, how does training the proposed projector $f_{\theta}$ compare against using a similar number of parameters to finetune the full backbone $g$ (e.g., via LoRA) on the E2E ELQ $\rightarrow$ HQ task? The paper seems to have created a new, more severe degradation setting  and then compared its method against baselines that were either not trained for this setting or were not fully trained for it.

REF:
@inproceedings{hsu2024drct,
  title={Drct: Saving image super-resolution away from information bottleneck},
  author={Hsu, Chih-Chung and Lee, Chia-Ming and Chou, Yi-Shiuan},
  booktitle={Proceedings of the IEEE/CVF Conference on Computer Vision and Pattern Recognition},
  pages={6133--6142},
  year={2024}
}
@article{zhu2024information,
  title={Information bottleneck based self-distillation: Boosting lightweight network for real-world super-resolution},
  author={Zhu, Han and Chen, Zhenzhong and Liu, Shan},
  journal={IEEE Transactions on Circuits and Systems for Video Technology},
  year={2024},
  publisher={IEEE}
}
@article{gao2025infobfr,
  title={Infobfr: Real-world blind face restoration via information bottleneck},
  author={Gao, Nan and Li, Jia and Huang, Huaibo and Shang, Ke and He, Ran},
  journal={arXiv preprint arXiv:2501.15443},
  year={2025}
}

inproceedings{relic2024lossy,
  title={Lossy image compression with foundation diffusion models},
  author={Relic, Lucas and Azevedo, Roberto and Gross, Markus and Schroers, Christopher},
  booktitle={European Conference on Computer Vision},
  pages={303--319},
  year={2024},
  organization={Springer}
}
@inproceedings{xu2025decouple,
  title={Decouple Distortion from Perception: Region Adaptive Diffusion for Extreme-low Bitrate Perception Image Compression},
  author={Xu, Jinchang and Wang, Shaokang and Chen, Jintao and Li, Zhe and Jia, Peidong and Zhao, Fei and Xiang, Guoqing and Hao, Zhijian and Zhang, Shanghang and Xie, Xiaodong},
  booktitle={Proceedings of the Computer Vision and Pattern Recognition Conference},
  pages={18051--18061},
  year={2025}
}

**Questions:**

1.The central claim of the paper hinges on the supposed inferiority of a direct ELQ $\rightarrow$ HQ mapping. To validate this, the authors must provide a comparison against a baseline diffusion model (e.g., OSEDiff) that has been fully trained from scratch on the same ELQ $\rightarrow$ HQ data, using a comparable parameter count and training budget. Why do the authors believe this direct mapping is "intractable," especially given that diffusion models routinely recover full images from pure noise?

2.The current method trains the projector $f_{\theta}$ while keeping the restorer $g$ frozen. This seems to be a practical choice. I am curious about the performance if the entire pipeline ($f_{\theta}$ and $g$) were trained jointly on the ELQ $\rightarrow$ HQ task. This experiment would help isolate whether the benefit comes from the structural decomposition (ELQ $\rightarrow$ LQ $\rightarrow$ HQ) itself or simply from the proposed pre-training strategy.

---

> ### Author Response · Authors · 2025-11-24
>
> **W1. Novelty regarding the use of the Information Bottleneck (IB) principle for image restoration needs to be more carefully justified and contextualized.**
>
> **A.** Our main novelty in using the IB principle is that prior works formulate the IB principle such that *the compressed representation Z are the features of a single restoration model*, while we formulate the IB principle such that *the HQ reconstruction itself is the compressed representation Z*. Under this new perspective, we model image restoration as the process of extracting relevant information from LQ for HQ reconstruction, and build a framework of multiple models for enhanced reconstruction.
>
> To elaborate, prior works build the Information Bottleneck chain $X\rightarrow Z \rightarrow Y$ such that: the input $X$ is taken as the LQ image, the output $Y$ as the HQ image, and the **intermediate restoration network features as $Z$**. The IB principle is introduced mainly as a tool to justify that the mutual information should be maximized.
>
> For instance, **DRCT [1]** proposed a new model design that introduces additional dense connections to overcome mutual information loss caused by the restoration network IB when training $LQ\rightarrow HQ$. While IB is mentioned as a phenomenon to overcome, it is used only as a conceptual explanation; DRCT doese not derive an explicit IB objective. Likewise, **InfoBFR [2]** introduced a manifold IB module to compress neural degradation and derive a relatively explicit objective. However, InfoBFR still views the restoration network itself as an IB chain, very different from our perspective.
>
> In our work, we build an IB chain $\hat{LQ} \rightarrow HQ \rightarrow \tilde{LQ}$, and view **HQ itself from an Information Bottleneck perspective and explicitly derive our loss from the IB objective.** This is possible because we factorized the more complex process of $ELQ \rightarrow HQ$ into two steps as $ELQ\rightarrow LQ\rightarrow HQ$.
>
>
> ---
>
> **W2, Q1. The paper's central premise is that a direct, end-to-end (E2E) mapping from ELQ to HQ is "challenging" or "intractable," thus justifying the proposed two-stage decomposition. However, the experimental evidence provided is insufficient to support this strong claim.**
>
> **A.** We would like to clarify that *the official implementation of OSEDiff tunes a diffusion network via LoRA fine-tuning of rank 4* to achieve their reported restoration capabilities. Thus, our baseline of ELQ→HQ training was performed in exactly the same manner as the official training of OSEDiff, with the only difference being degradation scale (changed from LQ settings to ELQ settings). To match these settings, we set the LoRA rank of our projector module to 4 as well, so the number of trainable parameters are essentially the same. Therefore, the comparisons we make are in fact performed under controlled settings and are quite fair.
>
>  However, we agree with the reviewer that a more extensive analysis could be provided regarding whether the ELQ → HQ task is in actually challenging for varying capabilities of the baseline model. Therefore, we provide thorough comparison between different LoRA ranks of 4, 16, 32, 64 for the baseline ELQ → HQ training in **Appendix B** of our updated manuscript. We mention that 64 is an unusually high rank for LoRA training, and greatly exceeds our original setting of rank 4. Increasing the LoRA rank in this way gives insight into the correlation between the number of trainable parameters and model performance.
>
> Importantly, we find that even if the LoRA rank increases, baseline performance does not increase; rather, a sharp decrease in PSNR is observed for the high LoRA rank of 64. Compared to baselines of all LoRA ranks, our method of ELQ→LQ→HQ training shows consistently better performance even with LoRA rank of 4.
>
> On a final note, we mention that the task of image restoration is considerably more ill-posed and unstable objective than unconditional noise-to-image generation, where the model only needs to sample from the prior p(x) without enforcing consistency with a given measurement. In the image restoration task, the model must simultaneously (i) undo an unknown degradation pipeline *and* (ii) recreate missing high-frequency content based on the prior p(x), all while remaining faithful to the ELQ input. The complexity of this task is highly correlated with the level of degradation, thus leading to our claim that the ELQ→HQ mapping is highly challenging and impractical to train. However, we do not claim that the ELQ→HQ mapping is fundamentally not learnable; given a very large model size, data, and resources, the mapping could be learned in the universal-approximation sense. We elaborate more on this topic in the next question regarding our motivation.
>
>
> ---

---

> ### Author Response · Authors · 2025-11-24
>
> **W3. The paper's motivation is unclear. It seems to oscillate between two different claims: a fundamental claim and and a practical claim.**
>
> **A.** We agree with the reviewer that, in principle, a sufficiently large conditional diffusion model trained with enough paired (ELQ, HQ) data and enough resources *could* learn the ELQ→HQ mapping. Our intention was not to claim that such a mapping is impossible in a universal-approximation sense. Thus, we have softened the wording around "intractable" in our updated manuscript to clarify our point about practicality.
>
> We clarify that our motivation is about the **practical difficulty** of training a *single* end-to-end model under the EBIR setting, where degradations are blind, heavily compounded, and strongly information-destroying. In this regime, the model must simultaneously (i) undo an unknown degradation pipeline and (ii) hallucinate missing high-frequency content, while remaining faithful to the ELQ input. This is a considerably ill-posed and unstable objective.
>
> We support this through a number of experiments. First, we examine whether the ELQ→HQ task is in actually challenging for varying capabilities of the baseline model in **Appendix B** of our updated manuscript. A thorough comparison among different LoRA ranks shows that our method shows improved performance even if a much higher number of parameters are trained in the baseline setting. We clarify again that the official implementation of OSEDiff tunes a diffusion network via LoRA fine-tuning of rank 4, so our original comparisons were indeed highly controlled and fair.
>
> We also provide an experiment comparing the baseline method and the proposed method with respect to the number of training iterations in **Figure 7** of our update manuscript. Our method achieves superior PSNR and FID with a much smaller number of training iterations, showing that our decomposition greatly increases both training speed *and* overall performance.
>
>
> ---
>
> **Q2. The current method keeps the restorer $g$ frozen. How would performance differ if the entire pipeline ($f_\theta$ and $g$) was trained jointly?**
>
> **A.** In addition to its practical advantages, our frozen restorer $g$ also contributes to overall training stability by keeping the LQ domain fixed. Consequently, if $g$ were to be jointly trained, the entire optimization process would likely become less stable, leading to degraded performance. Nonetheless, exploring the joint training of $f_\theta$ and $g$, as suggested, would be an interesting and promising direction for future research.
>
> ---
>
> References
>
> [1] DRCT: Saving Image Super-resolution away from Information Bottleneck, CVPRW 2024
>
> [2] InfoBFR: Real-World Blind Face Restoration via Information Bottleneck

---

### Official Review · Reviewer_RwKC · 2025-11-01

**Soundness:** 3
**Presentation:** 3
**Contribution:** 2
**Rating:** 4
**Confidence:** 4

**Summary:**

This paper proposes a novel framework for Extreme Blind Image Restoration (EBIR). By first projecting extremely low-quality (ELQ) images onto an intermediate low-quality (LQ) manifold and then restoring them with existing high-quality (HQ) models, the method decomposes the challenging ELQ-to-HQ mapping. The authors formulate the problem using the Information Bottleneck (IB) principle and design an IB-based loss function to stabilize training. They also introduce a “Look Forward Once (LFO)” prompt refinement strategy, and demonstrate the effectiveness of their approach through experiments and integration with existing restoration models.

**Strengths:**

1. The paper formalizes the extreme blind image restoration problem as an Information Bottleneck problem and derives a theoretically-driven loss function, providing a new perspective for the image restoration field.

2. The proposed ELQ→LQ→HQ decomposition framework effectively extends the applicability of existing restoration models without requiring fine-tuning of the main restoration model, making it easy to integrate and deploy in practice.

3. The paper conducts both quantitative and qualitative comparisons on multiple public datasets (DIV2K, DIV8K), covering mainstream methods, plug-and-play capability, and ablation studies, demonstrating the effectiveness and flexibility of the proposed approach.

**Weaknesses:**

1. Lack of innovation: Although applying the Information Bottleneck theory to image restoration is novel, the ELQ→LQ→HQ decomposition approach is quite common in existing two-stage or hierarchical EBIR frameworks. The main innovation lies in the theoretical interpretation and loss design.

2. Insufficient analysis of individual components in the IRIB loss (such as LQ reconstruction, HQ prior, and HQ fidelity). The effectiveness of the LFO strategy is only demonstrated with a limited number of iterations, lacking a more systematic analysis of different iteration counts.

3. The main experimental comparisons are with OSEDiff, Real-ESRGAN, SeeSR, and S3Diff, but lack experiments with the latest extreme restoration methods (such as DiffBIR and Chain-of-Zoom) to further validate the effectiveness of the method.

4. The improvements on pixel-level metrics (PSNR/SSIM) are limited, with the main advantages shown in perceptual metrics (LPIPS, FID, etc.), but the analysis of true semantic structure recovery is not detailed enough.

5. The paper does not provide information on the additional computational complexity introduced by the method (including FLOPS, parameter count, inference time, etc.). It is also unclear how much extra training time is required due to the added projection layer and new loss functions, which is important for evaluating the method's effectiveness.

**Questions:**

1. Could you provide independent ablation studies for each component of the IRIB loss to analyze their individual contributions to the final performance?

2. Could you provide information on the additional computational complexity introduced by the model (including FLOPS, parameter count, and inference time), as well as how much extra training time is required due to the added projection layer and new loss functions?

---

> ### Author Response · Authors · 2025-11-24
>
> **W1. Although applying the Information Bottleneck theory to image restoration is novel, the ELQ→LQ→HQ decomposition approach is quite common in existing two-stage or hierarchical EBIR frameworks.**
>
> **A.** Though many prior works decompose challenging problems into a set of easier sub-problems (e.g., DiffBIR [1]), our novelty lies in *what* main problem we decompose, and *how* the decomposition takes place.
>
> For example, in DiffBIR the main problem is to unify BIR, and this is solved by decomposing into two sub-problems: degradation removal and image regeneration. However, in our work, the main problem is to solve *extreme* BIR, which we define as restoring images degraded *beyond* the training settings of conventional models. This is a rarely studied problem, as the majority of work focuses more on better restoration *within* conventional training settings (e.g., the Real-ESRGAN [2] degradation pipeline). We bring a novel application of the Information Bottleneck theory to solve this under-explored problem, and propose a carefully designed loss under this theoretically grounded decomposition.
>
> In summary, contrary to the reviewer’s claim that the ELQ→LQ→HQ decomposition is common, we clarify that existing decomposition techniques mainly focus on a decomposing the LQ→HQ mapping. In this work, we introduce a new set of *extremely* low quality (ELQ) images and show how the highly intractable ELQ→HQ decomposition can be effectively solved by leveraging the LQ domain.
>
> ---
>
> **W2, Q1. Insufficient analysis of individual components in the IRIB loss (such as LQ reconstruction, HQ prior, and HQ fidelity).**
>
> **A.** Thanks for pointing this out. Below, we provide a thorough ablation study of each component in our final training objective. This table is also included as **Table 3** in our updated manuscript.
>
> **Ablation results on the DIV2K Dataset:**
> | $L_\text{LQ-recon}$ | $L_\text{HQ-prior}$ | $L_\text{HQ-fid}$ | PSNR ↑  | SSIM ↑ | LPIPS ↓ | DISTS ↓ | FID ↓    | NIQE ↓ | MUSIQ ↑ | CLIPIQA ↑ |
> |-----------|-----------|---------|---------|--------|---------|---------|----------|--------|---------|-----------|
> | ✗         | ✓         | ✓       | 19.2451 | 0.4592 | 0.4818  | 0.2801  | 71.5057  | 4.5954 | **72.7040** | 0.7303    |
> | ✓         | ✗         | ✓       | **20.0718** | **0.4898** | 0.4464  | 0.2808  | 75.1813  | 5.4442 | 70.3964 | **0.7449**    |
> | ✓         | ✓         | ✗       | 16.7439 | 0.3764 | 0.6740  | 0.3317  | 102.6394 | 4.3963 | 71.3685 | 0.6942    |
> | ✓         | ✓         | ✓       | 20.0488 | 0.4797 | **0.4249**  | **0.2621**  | **62.9768**  | **4.1814** | 70.3529 | 0.6829    |
>
> **Ablation results on the DIV8K Dataset:**
> | $L_\text{LQ-recon}$ | $L_\text{HQ-prior}$ | $L_\text{HQ-fid}$ | PSNR ↑  | SSIM ↑ | LPIPS ↓ | DISTS ↓ | FID ↓    | NIQE ↓ | MUSIQ ↑ | CLIPIQA ↑ |
> |-----------|-----------|---------|---------|--------|---------|---------|----------|--------|---------|-----------|
> | ✗         | ✓         | ✓       | 20.0605 | 0.4874 | 0.4583  | 0.2719  | 47.5777  | 4.7432 | **72.0902** | 0.7238    |
> | ✓         | ✗         | ✓       | **20.9121** | **0.5133** | 0.4290  | 0.2699  | 48.2483  | 5.4185 | 69.7655 | **0.7401**    |
> | ✓         | ✓         | ✗       | 17.5600 | 0.4126 | 0.6497  | 0.3230  | 77.3818  | 4.5927 | 70.2459 | 0.6955    |
> | ✓         | ✓         | ✓       | 20.8791 | 0.5040 | **0.4051**  | **0.2535**  | **40.7901**  | **4.3803** | 69.7483 | 0.6803    |
>
> The ablation results give sufficient insight into the purposes of each individual component. When either the $L_\text{LQ-recon}$ or $L_\text{HQ-fid}$ component is ablated, there is strict drop in reference-based consistency metrics (i.e., PSNR, SSIM, LPIPS, DISTS). Thus, these loss terms enable the model to generate images consistent with the ELQ input. When the $L_\text{HQ-prior}$ component is ablated, we observe degraded performance on metrics that evaluate perceptual quality (e.g., LPIPS, DISTS, FID, NIQE). Overall, our final objective provides an adequate balance between fidelity and perceptual quality.
>
> ---

---

> > ### Author Response · Authors · 2025-11-24
> >
> > **W2. The effectiveness of the LFO strategy is only demonstrated with a limited number of iterations, lacking a more systematic analysis of different iteration counts.**
> >
> > **A.** We empirically found that two LFO iterations are optimal in terms of the trade-off between computational cost and performance gain. To further substantiate this choice from an empirical perspective, we provide additional analysis of how FID saturates over LFO iterations (provided in **Figure 10**), where FID reaches its best value at the second iteration and then oscillates. This can be interpreted as suggesting that the maximum number of iterations from which we can extract meaningful text prompts is around 2 to 3.
> >
> > Additional qualitative results and examples are shown in **Figure 11**. Without LFO, the text that can be extracted under severe degradation is very limited, whereas as the iterations proceed, we confirm that meaningful and descriptive text prompts emerge to progressively refine the LQ image. Beyond 2 to 3 LFO iterations, the text prompt does not change meaningfully, as most objects in the image are successfully described. Thus, a saturation in performance is observed, which is supported by the FID saturation previously explained.
> >
> > ---
> >
> > **W3. Further comparison with the latest extreme restoration methods (ex. DiffBIR, Chain-of-Zoom) is needed.**
> >
> > **A.** Thanks for the suggestion. We give additional quantitative comparison with the latest extreme restoration methods DiffBIR [1] and SUPIR [3] in Table 1 of our updated manuscript. We do not compare with Chain-of-Zoom [4], as this work focuses on a completely different task of zooming into a local region of an image up to extreme scales, and is not a method for restoration. Furthermore, we remind the reviewer that extensive comparison with OSEDiff [5] is already given, and that OSEDiff is a state-of-the-art model proven to show much better performance than DiffBIR. Even compared to these additional image restoration methods, our method provides the best results for extreme blind image restoration.
> >
> > ---
> >
> > **W4. The improvements on pixel-level metrics (PSNR/SSIM) are limited, with the main advantages shown in perceptual metrics (LPIPS, FID, etc.), but the analysis of true semantic structure recovery is not detailed enough.**
> >
> > **A.** We remind the reviewer that the ablation study in **Appendix A** gives a thorough analysis regarding the tradeoff between fidelity and perceptual quality, dependent on the hyperparameter $\lambda_\text{blur}$. Various components of our final loss provide an adequate balance between pixel-level fidelity and perceptual quality, and users are able to freely balance fidelity and perceptual quality as preferred by adjusting $\lambda_\text{blur}$. Specifically, by setting $\lambda_\text{blur}$ to high values (e.g., $\lambda_\text{blur}$=2.0), we can produce results higher in both pixel-level metrics *and* perceptual metrics. The PSNR-MUSIQ and PSNR-FID curves in **Figure 8** visualize this trade-off, and support the superiority of our method.
> >
> > ---
> >
> > **W5, Q2. Could you provide information on the additional computational complexity introduced by the model, as well as how much extra training time is required due to the added projection layer and new loss functions?**
> >
> > **A.** Thank you for the suggestion. Details regarding computational complexity during inference time are summarized in **Appendix C** and **Table 5**. Furthermore, we provide training time analysis for different frameworks. **Figure 7** presents a comparison between the baseline model and our method with respect to the training iterations. With the same number of trainable parameters, our method more rapidly achieves superior performance in terms of fidelity (PSNR) and image quality (FID). Thus, our method *does not* incur additional training time; rather, it provides a cost-efficient method for high-quality reconstruction.
> >
> > ---
> > References
> >
> > [1] Diffbir: Toward blind image restoration with generative diffusion prior, ECCV 2024
> >
> > [2] Real-esrgan: Training real-world blind super-resolution with pure synthetic data, ICCV 2021
> >
> > [3] Scaling up to excellence: Practicing model scaling for photo-realistic image
> > restoration in the wild, CVPR 2024
> >
> > [4] Chain-of-zoom: Extreme super-resolution via scale autoregression and preference alignment, NeurIPS 2025
> >
> > [5] One-step effective diffusion network for real-world image super-resolution, NeurIPS 2024

---

> > > ### Comment · Reviewer_RwKC · 2025-11-28
> > > **feedback**
> > >
> > > Thank you for your thorough response, which has resolved some of my questions. However, there are still several issues that require clarification from the author. The answers to these questions will determine how many points I will award.
> > >
> > > 1. Both the LQ reconstruction term and HQ prior matching term in the loss function are simplifications of Equation 14. As I understand it, these include the author's own interpretations, such as those in line 264 and line 305. The forms of these losses differ from the Information Bottleneck for Extreme Image Restoration motivation mentioned in the paper, i.e., Equation 14. Is there any basis for these simplifications in the loss function? Are there other papers that support this approach?
> > >
> > > 2. The structure of the HQ prior matching term is similar to the VSD in OSEdiff, and the KL divergence part of the HQ prior matching term in Equation 14 is highly similar to the DMD method in the paper "One-step Diffusion with Distribution Matching Distillation". Why not use this form of loss? Instead, why is the definition in line 305 adopted?
> > >
> > > 3. In line 300, it is mentioned that the degradation generator D is Real-ESRGAN, which is quite confusing. The purpose of obtaining an intermediate LR frame is to reduce the learning difficulty. My understanding is that the simpler the degradation contained in this LR frame, the clearer the texture and semantic information it can provide. This intermediate LR is supervised by the GT-degraded estimated LR. Why not use a simple degradation method, such as bicubic? This would also reduce the reconstruction difficulty for the IR model. Furthermore, it is hard to judge whether the degradation distribution of ELQ is close to that of Real-ESRGAN. Real-ESRGAN is mainly used as an effective degradation method when real data is lacking, but its application in this paper seems inappropriate.
> > >
> > > 4. The multiple reuse of LFO can indeed produce denser text descriptions. However, when generating text descriptions from images, the inherent uncontrollability of the model may lead to descriptions of artifacts. This issue can be exacerbated by multiple reuse of LFO, as reflected in Table 1 (i.e., PSNR gradually decreases). However, super-resolution tasks require high fidelity in image generation, so the significance of the LFO part needs further discussion.
> > >
> > > 5. Using only OSEdiff as the basic framework lacks support from other baseline model results. If possible, it would be better to provide results under a new baseline. Given the limited rebuttal time, failure to do so will not affect my decision to lower the score.
> > >
> > > Thank you again for your response. I basically acknowledge the motivation and novelty of this paper, but I hope the author will seriously address the above details. The content of your reply will determine whether and how much I increase the score.

---

> > > > ### Author Response · Authors · 2025-12-03
> > > >
> > > > We sincerely thank the reviewer for the meaningful feedback, and we appreciate to hear that the motivation and novelty of our work has been recognized. Below, we address each question the reviewer has newly raised in detail:
> > > >
> > > > ---
> > > >
> > > > **Q1. Both the LQ reconstruction term and HQ prior matching term in the loss function are simplifications of Equation 14. Is there any basis for these simplifications in the loss function?**
> > > >
> > > > **A.**
> > > > In the preliminaries section of our original manuscript, we provide the $\beta$-VAE objective:
> > > >
> > > > $$
> > > > \mathcal{L_{\beta\text{-VAE}}}(\phi, \psi)
> > > > = \mathbb{E_{\text{p(x)}}} \Big[ \mathbb{E_{\text{q}\_\phi\text{(z|x)}}}\big[-\log q_\psi(x|z)\big]
> > > > \+\ \beta\ \mathrm{KL}\big(q_\phi(z|x)\\Vert\ r(z)\big) \Big].
> > > > $$
> > > >
> > > > which we can observe to be analogous to our proposed Eq. (14). Thus, in this work we view the posterior $q_\phi(z|x)$ as the information bottleneck for the reconstruction task that maximizes $\mathbb{E_{\text{q}\_\phi\text{(z|x)}}}[\log p(x|z)]$, allowing this objective to become a direct instance of the Information Bottleneck principle.
> > > >
> > > > In $\beta$-VAE [1], the first term (negative log-likelihood / reconstruction term) is instantiated as an MSE loss, and the KL divergence is computed in closed form under a Gaussian prior.
> > > >
> > > > In our framework, **we adopt these same simplifications as in $\beta$-VAE** into Eq. (14) :
> > > > - the **LQ reconstruction loss** corresponds to the negative log-likelihood term, and
> > > > - the **HQ prior matching term** corresponds to the KL term, $\mathrm{KL}(q_\phi(z|x)\\Vert\ r(z))$.
> > > >
> > > > The key difference is that our target prior $r(z) \approx p_{\text{HQ}}$ is an HQ diffusion prior rather than a simple Gaussian. Because the KL divergence to this HQ prior is intractable in closed form, we implement this term via a variational score distillation (VSD) loss, which serves as an approximation to reverse-KL distribution matching. This follows the same rationale as **ProlificDreamer [2]**, where VSD is explicitly used to approximate reverse-KL matching to a diffusion prior. To avoid any confusion, we have updated our manuscript to explicitly state that the practical loss is constructed *"under intuitions provided by previous work [1, 2]"*.
> > > >
> > > > ---
> > > >
> > > > **Q2. Could you provide the difference of the HQ prior term suggested in this paper and VSD in OSEDiff? The KL divergence term in Equation 14 seems highly similar to the DMD.**
> > > >
> > > > **A.** We thank the reviewer for this question and agree that our current writing may have caused confusion. Our HQ prior matching term is in fact based on the exact same variational score distillation (VSD) loss used in OSEDiff, combined with the standard diffusion loss for finetuning the regularizer.
> > > >
> > > > Concretely, our HQ prior matching loss is
> > > > $$
> > > > \mathcal{L_{\text{VSD}}}
> > > >     = \mathbb{E_{\text{x}, \text{z}\_\text{t}, \epsilon}}\Big[\tfrac{1}{2}\big\rVert\hat{\epsilon}(z_t, t) - \hat{\epsilon}_{\text{HQ}}(z_t, t)\big\rVert_2^2\Big]
> > > > $$
> > > > which corresponds to the same reverse-KL distribution matching objective that OSEDiff optimizes via VSD.
> > > >
> > > > $$
> > > > \mathcal{L_{\text{diff}}}
> > > >     = \mathbb{E}_{x, t, \epsilon}\Big[\tfrac{1}{2}\big\rVert\epsilon - \hat{\epsilon}(z_t, t)\big\rVert_2^2\Big],
> > > > $$
> > > >
> > > > Also following OSEDiff, the regularizer is optimized using $\mathcal{L_{\text{diff}}}$.
> > > > Thus, the final $\mathcal{L_{\text{HQ}\_prior}}$ becomes the sum of $\mathcal{L_{\text{VSD}}}$ and $\mathcal{L_{\text{diff}}}$ (up to weighting constants). We have updated the manuscript to explicitly clarify this connection to OSEDiff.

---

> > > > > ### Author Response · Authors · 2025-12-03
> > > > >
> > > > > **Q3-1. What is the purpose of applying the degradation generator from Real-ESRGAN, and not some other simple degradation (e.g., bicubic)?**
> > > > >
> > > > > **A** We thank the reviewer for this insightful question.
> > > > > Our original purpose of using Real-ESRGAN degradation was to allow for more robust training. Our hypothesis was that the broad distribution gap between "highly perturbed ELQ samples" and "bicubic-generated LQ samples" would make training $f_\theta$ difficult; thus, we originally chose "Real-ESRGAN perturbed LQ samples" in order to lessen the distribution gap and stabilize training.
> > > > >
> > > > > In response to the reviewer's question, we trained our method using "bicubic-generated LQ samples" and compared this setting against the original Real-ESRGAN–based setting on the DIV2K and DIV8K datasets:
> > > > >
> > > > > **Bicubic downsampling degradation results on the DIV2K Dataset:**
> > > > > | Degradation Method | LFO | PSNR ↑  | SSIM ↑ | LIPIS ↓ | DISTS ↓ | FID ↓   | NIQE ↓ | MUSIQ ↑ | CLIPIQA ↑ |
> > > > > |--------------------|-----|---------|--------|---------|---------|---------|--------|---------|-----------|
> > > > > | Real-ESRGAN        | x2  | 20.0047 | 0.4790 | **0.4255**  | 0.2594  | 62.9475 | **4.1875** | **70.6806** | **0.6848**    |
> > > > > | Bicubic (x4)       | x2  | **20.0686** | **0.4798** | 0.4267  | **0.2592**  | **62.1766** | 4.2285 | 70.3195 | 0.6810    |
> > > > >
> > > > > **Bicubic downsampling degradation results on the DIV8K Dataset:**
> > > > > | Degradation Method | LFO | PSNR ↑  | SSIM ↑ | LIPIS ↓ | DISTS ↓ | FID ↓   | NIQE ↓ | MUSIQ ↑ | CLIPIQA ↑ |
> > > > > |--------------------|-----|---------|--------|---------|---------|---------|--------|---------|-----------|
> > > > > | Real-ESRGAN        | x2  | 20.8421 | **0.5039** | **0.4053**  | **0.2505**  | 40.5523 | **4.3719** | **70.0020** | **0.6806**    |
> > > > > | Bicubic (x4)       | x2  | **20.8939** | 0.5034 | 0.4062  | 0.2517  | **40.4444** | 4.3981 | 69.8295 | 0.6789    |
> > > > >
> > > > > The results turned out to be comparable, showing that our method *does not* depend much on choice of degradation (*i.e.*, Real-ESRGAN or bicubic), and remains effective *regardless* of type of degradation used. Our interpretation into this is that: (1) bicubic downsampling is a *subset* of Real-ESRGAN degradation, thus the "bicubic-generated LQ samples" are not completely new, and (2) freezing the restoration model $g$ allows even simple degradations to work effectively. The results show that our method is highly robust, and does not rely on any typical setting of degradation. We thank the reviewer for helping us find this new advantage of our method, and acknowledge that a more systematic study along this axis would be an interesting direction for future research.
> > > > >
> > > > > Furthermore, we conducted plug-and-play experiments with SeeSR, which was originally trained under the Real-ESRGAN degradation pipeline. As shown in the tables below, the performance with bicubic downsampling is again quite similar to that with Real-ESRGAN. This suggests that, in practice, using a simple bicubic LQ degradation does not significantly harm compatibility with existing IR models trained with Real-ESRGAN, and our framework remains effective across both types of LQ distributions.
> > > > >
> > > > > **Plug and Play with SeeSR results on the DIV2K Dataset:**
> > > > > | Degradation Method | PSNR ↑  | SSIM ↑ | LIPIS ↓ | DISTS ↓ | FID ↓   | NIQE ↓ | MUSIQ ↑ | CLIPIQA ↑ |
> > > > > |--------------------|---------|--------|---------|---------|---------|--------|---------|-----------|
> > > > > | Real-ESRGAN        | 19.9312 | 0.4632 | 0.4506  | **0.2374**  | 61.2615 | 4.1355 | 71.5719 | **0.7320**    |
> > > > > | Bicubic (x4)       | **20.0200** | **0.4654** | **0.4492**  | 0.2376  | **60.8851** | **4.1257** | **71.6518** | 0.7284    |
> > > > >
> > > > > **Plug and Play with SeeSR results on the DIV8K Dataset:**
> > > > > | Degradation Method | PSNR ↑  | SSIM ↑ | LIPIS ↓ | DISTS ↓ | FID ↓   | NIQE ↓ | MUSIQ ↑ | CLIPIQA ↑ |
> > > > > |--------------------|---------|--------|---------|---------|---------|--------|---------|-----------|
> > > > > | Real-ESRGAN        | 20.8649 | **0.4921** | **0.4345**  | **0.2345**  | **39.6162** | 4.3091 | 70.7543 | 0.7285    |
> > > > > | Bicubic (x4)       | **20.9529** | 0.4920 | 0.4350  | 0.2364  | 39.8548 | **4.2648** | **70.7942** | **0.7295**    |

---

> > > > > > ### Author Response · Authors · 2025-12-03
> > > > > >
> > > > > > **Q3-2. It is hard to judge whether the ELQ distribution is close to that of Real-ESRGAN. Real-ESRGAN is mainly used as an effective degradation to simulate degraded data when lacking real data.**
> > > > > >
> > > > > > **A.**
> > > > > > The ELQ distribution we construct is not intended to closely match the original Real-ESRGAN distribution. Instead, our goal is to deliberately go *beyond* the typical degradation range where existing image restoration models (trained under Real-ESRGAN or similar settings) perform well, and to focus on out-of-distribution (OOD) cases of *extreme* degradation where these models tend to collapse.
> > > > > >
> > > > > > In particular, we are interested in robust restoration for images under extreme degradation, and not moderate resolution loss as for super-resolution problems. The ELQ setting is designed to include stronger noise, blur, and other complex artifacts, which better reflect the challenges seen in severely degraded real images. As shown in **Appendix E**, our method can successfully restore old photographs that suffer not only from low resolution but also from substantial noise and various degradation patterns, supporting the relevance of this broader ELQ regime.
> > > > > >
> > > > > > At the same time, as in prior work, we face a practical limitation: there is no large-scale paired dataset of extremely degraded real images and their clean counterparts. For this reason, we adopt Real-ESRGAN as a synthetic degradation generator to approximate a family of realistic degradations and then extend it to more severe settings to construct ELQ images. Thus, while we do not claim that the ELQ distribution is "close" to the original Real-ESRGAN distribution in a strict sense, our results on real old photos suggest that training under this extended synthetic regime is a useful proxy for handling highly degraded real-world images.
> > > > > >
> > > > > > ---
> > > > > >
> > > > > > **Q4. Multiple reuse of LFO can produce denser text description, but also generate artifacts where high fidelity is important in super-resolution task.**
> > > > > >
> > > > > > **A.** We appreciate the reviewer’s concern and we agree that repeatedly applying LFO can, in principle, slightly reduce pixel-level fidelity. In fact, as the number of LFO iterations increases, we do observe mild changes in pixel-wise measures. However, the PSNR and SSIM metrics are well known to favor overly smooth or blurred outputs, thus their reliance for small differences is questionable. On the other hand, Table 1 shows that perceptual fidelity metrics such as DISTS consistently improve as we increase the number of LFO iterations, indicating that the resulting images are perceptually closer to natural images and more semantically faithful.
> > > > > >
> > > > > > We would also like to remind the reviewer that extreme blind image restoration is fundamentally underdetermined. Perfectly reconstructing the original image is highly challenging, and even human experts would introduce a reasonable degree of subjectivity when restoring heavily degraded content. Thus, a small drop in pixel-level fidelity is reasonable if it leads to more realistic, image-likely reconstructions with improved FID and better perceptual quality.
> > > > > >
> > > > > > Finally, as we demonstrate in **Appendix 1**, we are able to *control* the balance between fidelity and perceptual quality in our framework. By appropriately choosing the number of LFO iterations and the weighting of the reconstruction term, we can operate in a high-fidelity regime where PSNR/SSIM remain high, while still outperforming baselines on most perceptual metrics. For example, the FID–PSNR trade-off curve shows that our method achieves a **strictly better Pareto frontier** compared to the baseline, suggesting that we can improve perceptual quality without any loss of fidelity, if fidelity is of more importance.
> > > > > >
> > > > > > ---
> > > > > >
> > > > > > **Q5. Are there any basic framework other than OSEDiff?**
> > > > > >
> > > > > > **A.** Through our experiments, we demonstrate that the best performance is achieved when the proposed ELQ$\rightarrow$LQ$\rightarrow$HQ training scheme is applied to the state-of-the-art baseline model, OSEDiff. We first report baseline results for a range of pretrained models to establish that OSEDiff outperforms other candidates, and then provide a detailed analysis showing how our proposed method further improves upon OSEDiff.
> > > > > >
> > > > > > Although we were unable to report results on additional IR backbones due to time and computational constraints, our extensive experiments already provide sufficient evidence for the soundness and robustness of the proposed approach. Extending our framework to other IR backbone models and systematically evaluating its generality is a promising direction for future work.
> > > > > >
> > > > > > ---
> > > > > >
> > > > > > References
> > > > > >
> > > > > > [1] $\beta$-VAE: Learning Basic Visual Concepts with a Constrained Variational Framework, ICLF 2017
> > > > > >
> > > > > > [2] ProlificDreamer: High-Fidelity and Diverse Text-to-3D Generation with Variational Score Distillation, NeurIPS 2023

---

### Official Review · Reviewer_ygqw · 2025-11-01

**Soundness:** 2
**Presentation:** 3
**Contribution:** 2
**Rating:** 6
**Confidence:** 2

**Summary:**

The paper proposes a new image restoration method designed for extreme degradations. The key idea is to learn a projector that maps an extremely degraded image onto an intermediate, less-degraded one before inferring the final high-quality image. The method demonstrates improved results when combined with an off-the-shelf image restoration (IR) approach.

**Strengths:**

1. The method generally shows improvements compared to the baseline or other compared methods.

2. Using the degraded image as a supervisory signal is interesting, although it was proposed before.

**Weaknesses:**

1. To me, the faithfulness with respect to the original input may be questionable.

**Questions:**

It is unclear what guarantees the consistency between the output and the input, particularly regarding the projector model’s ability to generate predictions that remain faithful to the input content.

---

> ### Author Response · Authors · 2025-11-24
>
> **W1, Q1. It is unclear what guarantees the consistency between the output and the input, particularly regarding the projector model’s ability to generate predictions that remain faithful to the input content.**
>
> **A.** In our final training objective of Eq. (19), the $L_\text{LQ-recon}$ and $L_\text{HQ-fid}$ terms effectively maintain the consistency between the output HQ reconstruction and the initial ELQ input. The specific details of each component $L_\text{LQ-recon}$ and $L_\text{HQ-fid}$ can be confirmed in Eq. (16) and Eq. (18), respectively. The gradients of these losses are backpropagated through the network up to the projector, effectively endowing the projector model the ability to generate predictions faithful to the input content.
>
> As further evidence, we additionally provide ablation study of each component in our final training objective. This table is also included as Table 3 in our updated manuscript.
>
> **Ablation results on the DIV2K Dataset:**
> | $L_\text{LQ-recon}$ | $L_\text{HQ-prior}$ | $L_\text{HQ-fid}$ | PSNR ↑  | SSIM ↑ | LPIPS ↓ | DISTS ↓ | FID ↓    | NIQE ↓ | MUSIQ ↑ | CLIPIQA ↑ |
> |-----------|-----------|---------|---------|--------|---------|---------|----------|--------|---------|-----------|
> | ✗         | ✓         | ✓       | 19.2451 | 0.4592 | 0.4818  | 0.2801  | 71.5057  | 4.5954 | **72.7040** | 0.7303    |
> | ✓         | ✗         | ✓       | **20.0718** | **0.4898** | 0.4464  | 0.2808  | 75.1813  | 5.4442 | 70.3964 | **0.7449**    |
> | ✓         | ✓         | ✗       | 16.7439 | 0.3764 | 0.6740  | 0.3317  | 102.6394 | 4.3963 | 71.3685 | 0.6942    |
> | ✓         | ✓         | ✓       | 20.0488 | 0.4797 | **0.4249**  | **0.2621**  | **62.9768**  | **4.1814** | 70.3529 | 0.6829    |
>
> **Ablation results on the DIV8K Dataset:**
> | $L_\text{LQ-recon}$ | $L_\text{HQ-prior}$ | $L_\text{HQ-fid}$ | PSNR ↑  | SSIM ↑ | LPIPS ↓ | DISTS ↓ | FID ↓    | NIQE ↓ | MUSIQ ↑ | CLIPIQA ↑ |
> |-----------|-----------|---------|---------|--------|---------|---------|----------|--------|---------|-----------|
> | ✗         | ✓         | ✓       | 20.0605 | 0.4874 | 0.4583  | 0.2719  | 47.5777  | 4.7432 | **72.0902** | 0.7238    |
> | ✓         | ✗         | ✓       | **20.9121** | **0.5133** | 0.4290  | 0.2699  | 48.2483  | 5.4185 | 69.7655 | **0.7401**    |
> | ✓         | ✓         | ✗       | 17.5600 | 0.4126 | 0.6497  | 0.3230  | 77.3818  | 4.5927 | 70.2459 | 0.6955    |
> | ✓         | ✓         | ✓       | 20.8791 | 0.5040 | **0.4051**  | **0.2535**  | **40.7901**  | **4.3803** | 69.7483 | 0.6803    |
>
> The ablation results above support our claims. When either the $L_\text{LQ-recon}$ or $L_\text{HQ-fid}$ component is ablated, there is strict drop in reference-based consistency metrics (i.e., PSNR, SSIM, LPIPS, DISTS). Thus, these loss terms enable the model to generate images consistent with the ELQ input.
>
> Finally, we have shown in **Appendix A** that users are able to freely balance fidelity and perceptual quality as preferred by adjusting the hyperparameter $\lambda_\text{blur}$. Thus, our framework not only guarantees the consistency between the output and the input, but actually gives the freedom to choose *how much* consistency is wanted.

---

### Meta-Review · Area_Chair_r3NJ · 2025-12-24

**Summary:**

This paper receives two marginally above the acceptance threshold and two marginally below the acceptance threshold. After the rebuttal, I think this paper still remains the following major issues: (1) novelty issue (RwKC augB), (2) insufficient experimental comparisons (RwKC augB J7vm) and (3) limited real-world validation (J7vm). As a result, this paper cannot be accepted in the current form.

**Reviewer Scores:**

NA

---

### Decision · Program_Chairs · 2026-01-26

Reject